# Every Parameter Matters: Ensuring the Convergence of Federated Learning with Dynamic Heterogeneous Models Reduction

**Hanhan Zhou**
The George Washington University
hanhan@gwu.edu

**Tian Lan**
The George Washington University
tlan@gwu.edu

**Guru Venkataramani**
The George Washington University
guruv@gwu.edu

**Wenbo Ding**
Tsinghua-Berkeley Shenzhen Institute
ding.wenbo@sz.tsinghua.edu.cn

## Abstract

Cross-device Federated Learning (FL) faces significant challenges where low-end clients that could potentially make unique contributions are excluded from training large models due to their resource bottlenecks. Recent research efforts have focused on model-heterogeneous FL, by extracting reduced-size models from the global model and applying them to local clients accordingly. Despite the empirical success, general theoretical guarantees of convergence on this method remain an open question. This paper presents a unifying framework for heterogeneous FL algorithms with online model extraction and provides a general convergence analysis for the first time. In particular, we prove that under certain sufficient conditions and for both IID and non-IID data, these algorithms converge to a stationary point of standard FL for general smooth cost functions. Moreover, we introduce the concept of minimum coverage index, together with model reduction noise, which will determine the convergence of heterogeneous federated learning, and therefore we advocate for a holistic approach that considers both factors to enhance the efficiency of heterogeneous federated learning.

## 1   Introduction

Federated Learning (FL) is a machine learning paradigm that enables a massive number of distributed clients to collaborate and train a centralized global model without exposing their local data [1]. Heterogeneous FL is confronted with two fundamental challenges: (1) mobile and edge devices that are equipped with drastically different computation and communication capabilities are becoming the dominant source for FL [2], also known as device heterogeneity; (2) state-of-the-art machine learning model sizes have grown significantly over the years, limiting the participation of certain devices in training. This has prompted significant recent attention to a family of FL algorithms relying on training reduced-size heterogeneous local models (often obtained through extracting a subnet or pruning a shared global model) for global aggregation. It includes algorithms such as HeteroFL [3] that employ fixed heterogeneous local models, as well as algorithms like PruneFL [4] and FedRolex [5] that adaptively select and train pruned or partial models dynamically during training. However, the success of these algorithms has only been demonstrated empirically (e.g., [2, 4, 3]). Unlike standard FL that has received rigorous analysis [6, 7, 8, 9], the convergence of heterogeneous FL algorithms is still an open question.

37th Conference on Neural Information Processing Systems (NeurIPS 2023).

This paper aims to answer the following questions: Given a heterogeneous FL algorithm that trains a shared global model through a sequence of time-varying and client-dependent local models, *what conditions can guarantee its convergence*? And intrinsically *how do the resulting models compare to that of standard FL*? There have been many existing efforts in establishing convergence guarantees for FL algorithms, such as the popular FedAvg [1], on both IID (independent and identically distributed data) and non-IID[9] data distributions, but all rely on the assumption that local models share the same uniform structure as the global model [1]. Training heterogeneous local models, which could change both over time and across clients in FL is desirable due to its ability to adapt to resource constraints and training outcomes[10].

For general smooth cost functions and under standard FL assumptions, we prove that heterogeneous FL algorithms satisfying certain sufficient conditions can indeed converge to a neighborhood of a stationary point of standard FL (with a small optimality gap that is characterized in our analysis), at a rate of $O(\frac{1}{\sqrt{Q}})$ in $Q$ communication rounds. Moreover, we show not only that FL algorithms involving local clients training different subnets (pruned or extracted from the global model) will converge, but also that the more they cover the parameters space in the global model, the faster the training will converge. Thus, local clients should be encouraged to train with reduced-size models that are of different subnets of the global model rather than pruning greedily. The work extends previous analysis on single-model adaptive pruning and subnetwork training[11, 12] to the FL context, where a fundamental challenge arises from FL's local update steps that cause heterogeneous local models (obtained by pruning the same global model or extracting a submodel) to diverge before the next aggregation. We prove

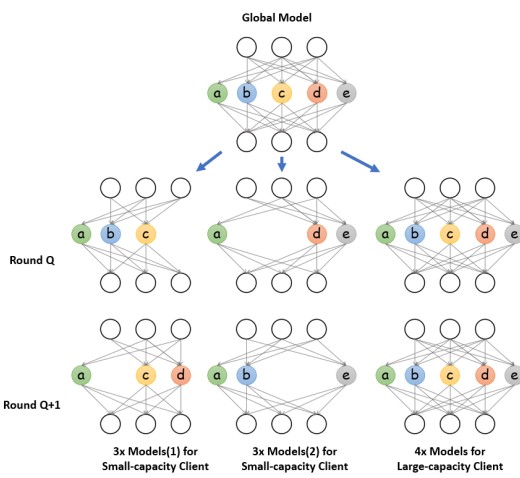

Figure 1: In this paper we show that instead of pruning small parameters greedily, local clients when applied with different local models not only will converge under certain conditions, it might even converge faster.

a new upperbound and show that the optimality gap (between heterogeneous and standard FL) is affected by both model-reduction noise and a new notion of minimum coverage index in FL (i.e., any parameters in the global model are included in at least $\Gamma_{\min}$ local models).

The key contribution of this paper is to establish convergence conditions for federated learning algorithms that employ heterogeneous arbitrarily-pruned, time-varying, and client-dependent local models to converge to a stationary point of standard FL. Numerical evaluations validate the sufficient conditions established in our analysis. The results demonstrate the benefit of designing new model reduction strategies with respect to both model reduction noise and minimum coverage index.

## 2 Background

**Standard Federated Learning** A standard FL problem considers a distributed optimization for N clients:

$$\min_{\theta} \left\{ F(\theta) \triangleq \sum_{i=1}^{N} p_i F_i(\theta) \right\}, \text{ with } F(\theta_i) = \mathbb{E}_{\xi \sim D_i} l(\xi_i, \theta_i), \tag{1}$$

where $\theta$ is as set of trainable weights/parameters, $F_n(\theta)$ is a cost function defined on data set $D_i$ with respect to a user specified loss function $l(x, \theta)$, and $p_i$ is the weight for the $i$-th client such that $p_i \geq 0$ and $\sum_{i=1}^{N} p_i = 1$.

---

[1]Throughout this paper, "non-IID data" means that the data among local clients are not independent and identically distributed. "Heterogeneous" means each client model obtained by model reduction from a global model can be different from the global model and other clients. "Dynamic" means time-varying, i.e. the model for one local client could change between each round.

The FL procedure, e.g., FedAvg [1], typically consists of a sequence of stochastic gradient descent steps performed distributedly on each local objective, followed by a central step collecting the workers' updated local parameters and computing an aggregated global parameter. For the $q$-th round of training, first, the central server broadcasts the latest global model parameters $\theta_q$ to clients $n = 1, \ldots, N$, who perform local updates as follows:

$$\theta_{q,n,t} = \theta_{q,n,t-1} - \gamma \nabla F_n(\theta_{q,n,t-1}; \xi_{n,t-1}) \text{ with } \theta_{q,n,0} = \theta_q$$

where $\gamma$ is the local learning rate. After all available clients have concluded their local updates (in $T$ epochs), the server will aggregate parameters from them and generate the new global model for the next round, i.e., $\theta_{q+1} = \sum_{n=1}^{N} p_i \theta_{q,n,T}$ The formulation captures FL with both IID and non-IID data distributions.

## 3 Related Work

**Federated Averaging and Related Convergence Analysis**. FedAvg [1] is consi dered the first and the most commonly used federated learning algorithm . Several works have shown the convergence of FedAvg under several different settings with both homogeneous (IID) data [6, 13] and heterogeneous (non-IID) data [9, 7, 8] even with partial clients participation [14]. Specifically, [8] demonstrated LocalSGD achieves $O(\frac{1}{\sqrt{NQ}})$ convergence for non-convex optimization and [9] established a convergence rate of $O(\frac{1}{Q})$ for strongly convex problems on FedAvg, where Q is the number of SGDs and N is the number of participated clients.

**Efficient and Heterogeneous FL through Neural Network Pruning and Sparsification.** Several works [15, 16, 17, 18, 19, 20] are proposed to further reduce communication costs in FL. One direction is to use data compression such as quantization [21, 7, 22, 23], sketching [24, 25], split learning [26], learning with gradient sparsity [27] and sending the parameters selectively[28]. This type of work does not consider computation efficiency. There are also works that address the reduction of both computation and communication costs, including one way to utilize lossy compression and dropout techniques[29, 30]. Although early works mainly assume that all local models share the same architecture as the global model [31], recent works have empirically demonstrated that federated learning with heterogeneous client models to save both computation and communication is feasible. PruneFL[4] proposed an approach with adaptive parameter pruning during FL. [32] proposed FL with a personalized and structured sparse mask. FjORD[33] and HetroFL[3] proposed to generate heterogeneous local models as a subnet of the global network by extracting a static sub-models, Hermes[34] finds the small sub-network by applying the structured pruning. There are also researches on extracting a subnetwork dynamically, e.g. Federated Dropout[29] extracts submodels randomly and FedRolex[5] applies a rolling sub-model extraction.

Despite their empirical success, they either lack theoretical convergence analysis or are specific to their own work. PruneFL only shows a convergence of the proposed algorithm and does not ensure convergence to a solution of standard FL. Meanwhile, static subnet extraction like Hermes does not allow the pruned local networks to change over time nor develop general convergence conditions. Following Theorem 1 works like Hermes can now employ time-varying subnet extractions, rather than static subnets, while still guaranteeing the convergence to standard FL. The convergence of HeteroFL and FedRolex– which was not available – now follows directly from Theorem 1. Once PruneFL satisfies the conditions established in our Theorem 1, convergence to a solution of standard FL can be achieved, rather than simply converging to some point. In summary, our general convergence conditions in Theorem 1 can provide support to existing FL algorithms that employ heterogeneous local models, ensuring convergence to standard FL. It also enables the design of optimized pruning masks/models to improve the minimum coverage index and thus the resulting gap to standard FL.

## 4 Methodology

### 4.1 Problem Formulation for FL with Heterogeneous Local models

Given an FL algorithm that trains heterogeneous local models for global aggregation, our goal is to analyze its convergence with respect to a stationary point of standard FL. We consider a general formulation where the heterogeneous local models can be obtained using any model reduction

strategies that are both (i) time-varying to enable online adjustment of reduced local models during the entire training process and (ii) different across FL clients with respect to their individual heterogeneous computing resource and network conditions. More formally, we denote the sequence of local models used by a heterogeneous FL algorithm by masks $m_{q,n} \in \{0,1\}^{|\theta|}$, which can vary at any round $q$ and for any client $n$. Let $\theta_q$ denote the global model at the beginning of round $q$ and $\odot$ be the element-wise product. Thus, $\theta_q \odot m_{q,n}$ defines the trainable parameters of the reduced local model[2] for client $n$ in round $q$. Our goal is to find sufficient conditions on such masks $m_{q,n} \; \forall q, n$ for the convergence of heterogeneous FL.

Here, we describe one around (say the $q$th) of the heterogeneous FL algorithm. First, the central server employs a given model reduction strategy $\mathbb{P}(\cdot)$ to reduce the latest global model $\theta_q$ and broadcast the resulting local models to clients:

$$\theta_{q,n,0} = \theta_q \cdot m_{q,n}, \text{ with } m_{q,n} = \mathbb{P}(\theta_q, n, q), \; \forall n. \tag{2}$$

We note that the model reduction strategy $\mathbb{P}(\theta_q, n, q)$ can vary over time $q$ and across clients $n$ in heterogeneous FL. Each client $n$ then trains the reduced local model by performing $T$ local updates (in $T$ epochs):

$$\theta_{q,n,t} = \theta_{q,n,t-1} - \gamma \nabla F_n(\theta_{q,n,t-1}, \xi_{n,t-1}) \odot m_{q,n}, \text{for } t = 1 \ldots T,$$

where $\gamma$ is the learning rate and $\xi_{n,t-1}$ are independent samples uniformly drawn from local data $D_n$ at client $n$. We note that $\nabla F_n(\theta_{q,n,t-1}, \xi_{n,t-1}) \odot m_{q,n}$ is a local stochastic gradient evaluated using only local parameters in $\theta_{q,n,t-1}$ (available to the heterogeneous local model) and that only locally trainable parameters are updated by the stochastic gradient (via an element-wise product with $m_{q,n}$).

Finally, the central server aggregates the local models $\theta_{n,q,T} \; \forall n$ and produces an updated global model $\theta_{q+1}$. Due to the use of heterogeneous local models, each global parameter is included in a (potentially) different subset of the local models. Let $\mathcal{N}_q^{(i)}$ be the set of clients, whose local models contain the $i$th modeling parameter in round $q$. That is $n \in \mathcal{N}_q^{(i)}$ if $m_{q,n}^{(i)} = 1$ and $n \notin \mathcal{N}_q^{(i)}$ if $m_{q,n}^{(i)} = 0$. Global update of the $i$th parameter is performed by aggregating local models with the parameter available, i.e.,

$$\theta_{q+1}^{(i)} = \frac{1}{|\mathcal{N}_q^{(i)}|} \sum_{n \in \mathcal{N}_q^{(i)}} \theta_{q,n,T}^{(i)}, \; \forall i, \tag{3}$$

where $|\mathcal{N}_q^{(i)}|$ is the number of local models containing the $i$th parameter. We summarize the algorithm details in Algorithm 1 and the explanation of the notations in the Appendix.

The fundamental challenge of convergence analysis mainly stems from the local updates in Eq.(3). While the heterogeneous models $\theta_{q,n,0}$ provided to local clients at the beginning of round $q$ are obtained from the same global model $\theta_q$, performing $T$ local updates causes these heterogeneous local models to diverge before the next aggregation. In addition, each parameter is (potentially) aggregated over a different subset of local models in Eq.(3). These make existing convergence analysis intended for single-model adaptive pruning [11, 12] non-applicable to heterogeneous FL. The impact of local model divergence and the global aggregation of heterogeneous models must be characterized in order to establish convergence.

The formulation proposed above captures heterogeneous FL with any model pruning or sub-model extraction strategies since the resulting masks $m_{q,n} \; \forall q, n$ can change over time $q$ and across clients $n$. It incorporates many model reduction strategies (such as pruning, sparsification, and sub-model extraction) into heterogeneous FL, allowing the convergence results to be broadly applicable. It recovers many important FL algorithms recently proposed, including HeteroFL [3] that uses fixed masks $m_n$, PruneFL [4] that periodically trains a full-size local model with $m_{n,q} = 1$, Prune-and-Grow [12] that can be viewed as a single-client version, as well as FedAvg [1] that employs full-size local models with $m_{n,q} = 1$ at all clients. Our analysis establishes general conditions for *any* heterogeneous FL with arbitrary online model reduction that may vary over communication rounds to converge to standard FL.

---

[2]While a reduced local model has a smaller number of parameters than the global model. We adopt the notations in [4, 35, 12] and use $\theta_q \odot m_{q,n}$ with an element-wise product to denote the pruned local model or the extracted submodel - only parameter corresponding to a 1-value in the mask is accessible and trainable in the local model.

Our paper establishes convergence conditions in the general form, which apply to both static and dynamically changing masks. With dynamically changing masks, we denote the reduced models/networks as $\theta_{q,n,t}$, which means that the model structure (with its corresponding mask) can change between each round of communications and during each local training epoch, and can be different from other local clients. We show that as long as the dynamic heterogenous FL framework can be framed as the setting above, our convergence analysis in this paper applies.

## 4.2 Notations and Assumptions

We make the following assumptions that are routinely employed in FL convergence analysis. In particular, Assumption 1 is a standard and common setting assuming Lipschitz continuous gradients. Assumption 2 follows from [12] (which is for a single-worker case) and implies the noise introduced by model reduction is bounded and quantified. This assumption is required for heterogeneous FL to converge to a stationary point of standard FL. Assumptions 3 and 4 are standard for FL convergence analysis following from [36, 37, 8, 9] and assume the stochastic gradients to be bounded and unbiased.

**Assumption 1.** *(Smoothness). Cost functions $F_1, \ldots, F_N$ are all L-smooth: $\forall \theta, \phi \in \mathcal{R}^d$ and any $n$, we assume that there exists $L > 0$:*

$$\|\nabla F_n(\theta) - \nabla F_n(\phi)\| \leq L\|\theta - \phi\|. \tag{4}$$

**Assumption 2.** *(Model Reduction Noise). We assume that for some $\delta^2 \in [0, 1)$ and any $q, n$, the model reduction error is bounded by*

$$\|\theta_q - \theta_q \odot m_{q,n}\|^2 \leq \delta^2 \|\theta_q\|^2. \tag{5}$$

**Assumption 3.** *(Bounded Gradient). The expected squared norm of stochastic gradients is bounded uniformly, i.e., for constant $G > 0$ and any $n, q, t$:*

$$\mathbb{E}_{\xi_{q,n,t}} \|\nabla F_n(\theta_{q,n,t}, \xi_{q,n,t})\|^2 \leq G. \tag{6}$$

**Assumption 4.** *(Gradient Noise for IID data). Under IID data distribution, $\forall q, n, t$, we assume a gradient estimate with bounded variance:*

$$\mathbb{E}_{\xi_{n,t}} \|\nabla F_n(\theta_{q,n,t}, \xi_{n,t}) - \nabla F(\theta_{q,n,t})\|^2 \leq \sigma^2 \tag{7}$$

## 4.3 Convergence Analysis

We now analyze the convergence of heterogeneous FL for general smooth cost functions. We begin with introducing a new notion of **minimum covering index**, defined in this paper by

$$\Gamma_{\min} = \min_{q,i} |\mathcal{N}_q^{(i)}|, \tag{8}$$

where $\Gamma_{\min}$ [3] measures the minimum occurrence of the parameter in the local models in all rounds, considering $|\mathcal{N}_q^{(i)}|$ is the number of heterogeneous local models containing the $i$th parameter. Intuitively, if a parameter is never included in any local models, it is impossible to update it. Thus conditions based on the covering index would be necessary for the convergence toward standard FL (with the same global model). All proofs for theorems and lemmas are collected in the Appendix with a brief proof outline provided here.

**Theorem 1.** *Under Assumptions 1-4 and for arbitrary masks satisfying $\Gamma_{\min} \geq 1$, when choosing $\gamma \leq 1/(6LT) \wedge \gamma \leq 1/(T\sqrt{Q})$, heterogeneous FL converges to a small neighborhood of a stationary point of standard FL as follows:*

$$\frac{1}{Q} \sum_{q=1}^{Q} \mathbb{E}\|\nabla F(\theta_q)\|^2 \leq \frac{G_0}{\sqrt{Q}} + \frac{V_0}{T\sqrt{Q}} + \frac{H_0}{Q} + \frac{I_0}{\Gamma^*} \cdot \frac{1}{Q} \sum_{q=1}^{Q} \mathbb{E}\|\theta_q\|^2$$

*where $G_0 = 4\mathbb{E}[F(\theta_0)]$, $V_0 = 6LN\sigma^2/(\Gamma^*)^2$, $H_0 = 2L^2NG/\Gamma^*$, and $I_0 = 3L^2\delta^2N$ are constants depending on the initial model parameters and the gradient noise.*

---

[3]We refer to $\Gamma^*$ for all equations and derivations.

An obvious case here is that when $\Gamma_{min} = 0$, where there exists at least one parameter that is not covered by any of the local clients and all the client models can not cover the entire global model, we can consider the union of all local model parameters, the *"largest common model"* among them, as a new equivalent global model $\hat{\theta}$ (which have a smaller size than $\theta$). Then, each parameter in $\hat{\theta}$ is covered in at least one local model. Thus Theorem 1 holds for $\hat{\theta}$ instead and the convergence is proven – to a stationary point of $\hat{\theta}$ rather than $\theta$.[4]

**Assumption 5.** *(Gradient Noise for non-IID data). Let $\hat{g}_{q,t}^{(i)} = \frac{1}{|\mathcal{N}_q^{(i)}|} \sum_{n \in \mathcal{N}_q^{(i)}} \nabla F_n^{(i)}(\theta_{q,n,t}, \xi_{n,t})$. Under non-IID data distribution, we assume $\forall i, q, t$ a gradient estimate with bounded variance:*

$$\mathbb{E}_\xi \left\| \hat{g}_{q,n,t}^{(i)} - \nabla F^{(i)}(\theta_{q,n,t}) \right\|^2 \le \sigma^2.$$

**Theorem 2.** *Under Assumptions 1-3 and 5, heterogeneous FL satisfying $\Gamma_{\min} \ge 1$, when choosing $\gamma \le 1/\sqrt{TQ}$ and $\gamma \le 1/(6LT)$, heterogeneous FL converges to a small neighborhood of a stationary point of standard FL as follows:*

$$\frac{1}{Q} \sum_{q=1}^{Q} \mathbb{E}\|\nabla F(\theta_q)\|^2 \le \frac{G_1}{\sqrt{TQ}} + \frac{V_0}{\sqrt{Q}} + \frac{I_0}{\Gamma^*} \cdot \frac{1}{Q} \sum_{q=1}^{Q} \mathbb{E}\|\theta_q\|^2 \tag{9}$$

*where $G_1 = 4\mathbb{E}[F(\theta_0)] + 6LK\sigma^2$.*

**Proof outline.** There are a number of challenges in delivering main theorems. We begin the proof by analyzing the change of loss function in one round as the model goes from $\theta_q$ to $\theta_{q+1}$, i.e., $F(\theta_{q+1}) - F(\theta_1)$. It includes three major steps: reducing the global model to obtain heterogeneous local models $\theta_{q,n,0} = \theta_q \odot m_{q,n}$, training local models in a distributed fashion to update $\theta_{q,n,t}$, and parameter aggregation to update the global model $\theta_{q+1}$.

Due to the use of heterogeneous local models whose masks $m_{q,n}$ both vary over rounds and change for different workers, we first characterize the difference between local model $\theta_{q,n,t}$ at any epoch $t$ and global model $\theta_q$ at the beginning of the current round. It is easy to see that this can be factorized into two parts: model reduction error $\|\theta_{q,n,0} - \theta_q\|^2$ and local training $\|\theta_{q,n,t} - \theta_{q,n,0}\|^2$, which will be analyzed in Lemma 1.

**Lemma 1.** *Under Assumption 2 and Assumption 3, for any q, we have:*

$$\sum_{t=1}^{T} \sum_{n=1}^{N} \mathbb{E}\|\theta_{q,n,t-1} - \theta_q\|^2 \le \gamma^2 T^2 NG + \delta^2 NT \cdot \mathbb{E}\|\theta_q\|^2 \tag{10}$$

We characterize the impact of heterogeneous local models on global parameter updates. Specifically, we use an ideal local gradient $\nabla F_n(\theta_q)$ as a reference point and quantify the difference between aggregated local gradients and the ideal gradient. This will be presented in Lemma 2.

**Lemma 2.** *Under Assumptions 1-3, for any q, we have:*

$$\sum_{i=1}^{K} \mathbb{E}\| \frac{1}{\Gamma_q^{(i)} T} \sum_{t=1}^{T} \sum_{n \in \mathcal{N}_q^{(i)}} [\nabla F_n^{(i)}(\theta_{q,n,t-1}) - \nabla F_n^{(i)}(\theta_q)]\|^2 \le \frac{L^2 \gamma^2 TNG}{\Gamma^*} + \frac{L^2 \delta^2 N}{\Gamma^*} \mathbb{E}\|\theta_q\|^2,$$

where we relax the inequality by choosing the smallest $\Gamma^* = \min_{q,i} \Gamma_q^{(i)}$. We also quantify the norm difference between a gradient and a stochastic gradient (with respect to the global update step) using the gradient noise assumptions, in Lemma 3.

---

[4]To better illustrate this scenario of $\Gamma_{min} = 0$, we will introduce an illustrative simplified example as follows: A global model $\theta = <\theta_1, \theta_2, \theta_3>$ where there will be two local models $\theta_a = <\theta_1>$ and $\theta_b = <\theta_3>$. Although $\Gamma_{min} = 0$ regarding the global model $\theta$, but for their largest common model, the union of $\theta_a$ and $\theta_b$ which is $<\theta_1, \theta_3>$ will become the new conceptual global model $\hat{\theta}$, where $\Gamma_{min} = 1$ regarding this conceptual global model. Thus the convergence still stands, but it will converge to a stationary point of FL with a different global model.

Since IID and non-IID data distributions in our model differ in the gradient noise assumption (i.e., Assumption 4 and Assumption 5), we present a unified proof for both cases. We will explicitly state IID and non-IID data distributions only if the two cases require different treatments (when the gradient noise assumptions are needed). Otherwise, the derivations and proofs are identical for both cases.

**Lemma 3.** *For IID data distribution under Assumptions 4, for any q, we have:*

$$\sum_{i=1}^{K} \mathbb{E}\|\frac{1}{\Gamma_q^{(i)}T}\sum_{t=1}^{T}\sum_{n\in\mathcal{N}_q^{(i)}} \nabla F_n^{(i)}(\theta_{q,n,t-1},\xi_{n,t-1}) - \nabla F^{(i)}(\theta_{q,n,t-1})\|^2 \leq \frac{N\sigma^2}{T(\Gamma^*)^2}$$

*For non-IID data distribution under Assumption 5, for any q, we have:*

$$\sum_{i=1}^{K} \mathbb{E}\|\frac{1}{\Gamma_q^{(i)}T}\sum_{t=1}^{T}\sum_{n\in\mathcal{N}_q^{(i)}} \nabla F_n^{(i)}(\theta_{q,n,t-1},\xi_{n,t-1}) - \nabla F^{(i)}(\theta_{q,n,t-1})\|^2 \leq \frac{K\sigma^2}{T}$$

Finally, under assumption 1, we have $F(\theta_{q+1}) - F(\theta_q) \leq \langle \nabla F(\theta_q), \ \theta_{q+1} - \theta_q \rangle + \frac{L}{2}\|\theta_{q+1} - \theta_q\|^2$ and use the preceding lemmas to obtain two upperbounds for the two terms. Combining these results we prove the desired convergence result in theorem 1 and theorem 2.

Theorem 1 shows the convergence of heterogenous FL to a neighborhood of a stationary point of standard FL albeit a small optimality gap due to model reduction noise, as long as $\Gamma_{\min} \geq 1$. The result is a bit surprising since $\Gamma_{\min} \geq 1$ only requires each parameter to be included in at least one local model – which is obviously necessary for all parameters to be updated during training. But we show that this is also a sufficient condition for convergence. Moreover, we also establish a convergence rate of $O(\frac{1}{\sqrt{Q}})$ for arbitrary model reduction strategies satisfying the condition. When the cost function is strongly convex (e.g., for softmax classifier, logistic regression, and linear regression with $l_2$-normalization), the stationary point becomes the global optimum. Thus, Theorem 1 shows convergence to a small neighborhood of the global optimum of standard FL for strongly convex cost functions.

## 5 Interpreting and Applying the Unified Framework

**Discussion on the Impact of model reduction noise.** In Assumption 2, we assume the model reduction noise is relatively small and bounded with respect to the global model: $\|\theta_q - \theta_q \odot m_{q,n}\|^2 \leq \delta^2 \|\theta_q\|^2$. This is satisfied in practice since most pruning strategies tend to focus on eliminating weights/neurons that are insignificant, therefore keeping $\delta^2$ indeed small. We note that similar observations are made on the convergence of single-model adaptive pruning [11, 12], but the analysis does not extend to FL problems where the fundamental challenge comes from local updates causing heterogeneous local models to diverge before the next global aggregation. We note that for heterogeneous FL, reducing a model will incur an optimality gap $\delta^2 \frac{1}{Q}\sum_{q=1}^{Q}\mathbb{E}\|\theta_q\|^2$ in our convergence analysis, which is proportional to $\delta^2$ and the average model norm (averaged over $Q$). It implies that a more aggressive model reduction in heterogeneous FL may lead to a larger error, deviating from standard FL at a speed quantified by $\delta^2$. We note that this error is affected by both $\delta^2$ and $\Gamma_{\min}$.

**Discussion on the Impact of minimum covering index $\Gamma_{\min}$.** The minimum number of occurrences of any parameter in the local models is another key factor in deciding convergence in heterogeneous FL. As $\Gamma_{\min}$ increases, both constants $G_0, V_0$, and the optimality gap decrease. Recall that our analysis shows the convergence of all parameters in $\theta_q$ with respect to a stationary point of standard FL (rather than for a subset of parameters or to a random point). The more times a parameter is covered by local models, the sooner it gets updated and convergences to the desired target. This is quantified in our analysis by showing that the optimality gap due to model reduction noise decreases at the rate of $\Gamma_{\min}$.

**Connections between model reduction noise and minimum covering index.** In this paper, we introduced the concept of minimum coverage index for the first time, where we show that only model compression alone is not enough to allow a unified convergence analysis/framework for

heterogeneous federated learning. The minimum coverage index, together with pruning/compression noises, determines convergence in heterogeneous FL. Our results show that heterogeneous FL algorithms satisfying certain sufficient conditions can indeed converge to a neighborhood of a stationary point of standard FL. This is a stronger result as it shows convergence to standard FL, rather than simply converging somewhere. A minimum coverage index of $\Gamma_{min} = 0$ means that the model would never be updated, which is meaningless even if it still converges.

**Discussion for non-IID case.** We note that Assumption 5 is required to show convergence with respect to standard FL and general convergence may reply on weaker conditions. We also notice that $\Gamma_{\min}$ no longer plays a role in the optimality gap. This is because the stochastic gradients computed by different clients in $\mathcal{N}_q^{(i)}$ now are based on different datasets and jointly provide an unbiased estimate, no longer resulting in smaller statistical noise.

**Applying the main theoretical findings.** Theorem 1 also inspires new design criteria for designing adaptive model-reducing strategies in heterogeneous FL. Since the optimality gap is affected by both model-reduction noise $\delta^2$ and minimum covering index $\Gamma_{\min}$, we may prefer strategies with small $\delta^2$ and large $\Gamma_{\min}$, in order to minimize the optimality gap to standard FL.

The example shown in Figure 1 illustrates alternative model reduction strategies in heterogeneous FL for $N = 10$ clients. Suppose all 6 low-capacities clients are using the reduced-size model by pruning greedily, which covers the same region of the global model, scenarios like this will only produce a maximum $\Gamma_{min} = 4$; however when applying low-capacities local clients with models covering different regions of the global model, $\Gamma_{min}$ can be increased, as an example we show how to design local models so $\Gamma_{min}$ is increased to 7 without increasing any computation and communication cost. The optimal strategy corresponds to lower noise $\delta^2$ while reaching a higher covering index. Using these insights, We present numerical examples with optimized designs in Section 6.

# 6 Experiments

## 6.1 Experiment settings

In this section, we evaluate heterogeneous FL with different model reduction strategies and aim to validate our theory. We focus on two key points in our experiments: (i) whether heterogeneous FL will converge with different local models and (ii) the impacts of key factors to the FL performances including minimum coverage index $\Gamma_{\min}$ and model-reduction noise $\delta^2$.

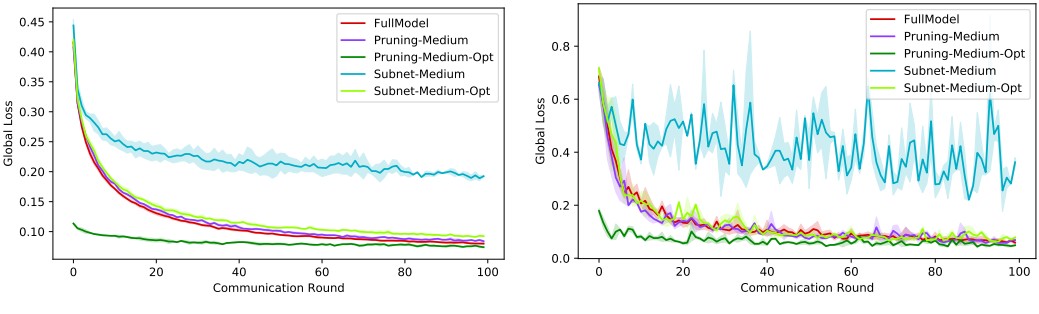

     (a) MLP trained on MNIST with IID data         (b) MLP trained on MNIST with non-IID data

Figure 2: Selected experimental results for MNIST with IID (a) and Non-IID (b) with high data heterogeneity data on medium model reduction level. "Opt" stands for optimized local model distribution covering more regions for a higher $\Gamma_{min}$, others do pruning greedily. As the shallow MLP is already at a small size, applying a medium level of model reduction will bring a high model reduction loss for subnet extraction method.

**Datasets and Models.** We examine the theoretical results on the following three commonly-used image classification datasets: MNIST [38] with a shallow multilayer perception (MLP), CIFAR-10 with Wide ResNet28x2 [39], and CIFAR100 [40] with Wide ResNet28x8[39]. The first setting where using MLP models is closer to the theoretical assumptions and settings, and the latter two settings are closer to the real-world application scenarios. We prepare $N = 100$ workers with IID and non-IID

| Model Reduction Level | Model Setting | $\Gamma_{min}$ | MNIST | | | CIFAR-10 | | | CIFAR100 | | |
|---|---|---|---|---|---|---|---|---|---|---|---|
| | | | IID | Non-IID | | IID | Non-IID | | IID | Non-IID | |
| | | | | L=5 | L=2 | | L=5 | L=2 | | L=50 | L=20 |
| FullModel | Homogenous-Full | 10 | 98.08 | 97.70 | 93.59 | 70.63 | 65.12 | 61.08 | 67.34 | 66.74 | 64.38 |
| | Pruning-Greedy | 6 | 98.18 | 97.60 | 93.15 | 72.66 | 62.49 | 57.17 | 67.41 | 65.67 | 65.06 |
| Low | Pruning-Optimised | 8 | 98.53 | 98.25 | 95.85 | 76.26 | 66.25 | 59.98 | 67.47 | 66.56 | 67.47 |
| Model | Static Subnet Subtraction | 6 | 97.62 | 95.12 | 92.33 | 73.20 | 61.25 | 56.09 | 66.60 | 65.24 | 65.79 |
| Reduction | Subnet Subtraction - Optimised | 8 | 97.76 | 94.41 | 93.60 | 73.78 | 64.17 | 58.09 | 67.81 | 66.66 | 67.23 |
| | Homogenous-Large | 10 | 97.52 | 96.08 | 93.23 | 69.05 | 63.72 | 57.42 | 66.81 | 65.57 | 63.90 |
| | Pruning-Greedy | 4 | 97.51 | 95.05 | 91.86 | 66.85 | 60.93 | 56.98 | 52.92 | 45.88 | 45.68 |
| Medium | Pruning-Optimised | 8 | 98.39 | 98.02 | 95.48 | 71.43 | 66.94 | 56.93 | 55.32 | 46.81 | 45.74 |
| Model | Static Subnet Subtraction | 4 | 95.56 | 92.33 | 92.05 | 61.87 | 58.08 | 46.03 | 50.59 | 44.22 | 45.25 |
| Reduction | Subnet Subtraction - Optimised | 8 | 97.96 | 94.05 | 93.36 | 63.96 | 62.65 | 47.44 | 52.95 | 46.23 | 46.15 |
| | Homogenous-Medium | 10 | 97.05 | 92.71 | 90.82 | 59.21 | 57.61 | 53.43 | 52.19 | 36.08 | 34.06 |
| | Pruning-Greedy | 3 | 95.01 | 86.83 | 76.64 | 67.35 | 56.75 | 22.55 | 39.29 | 26.14 | 25.97 |
| High | Pruning-Optimised | 5 | 95.32 | 91.98 | 81.66 | 67.74 | 57.33 | 27.97 | 40.78 | 29.63 | 26.63 |
| Model | Static Subnet Subtraction | 3 | 95.88 | 81.64 | 71.64 | 68.78 | 56.88 | 30.61 | 41.18 | 27.55 | 26.23 |
| Reduction | Subnet Subtraction - Optimised | 5 | 94.41 | 90.70 | 85.82 | 69.15 | 57.98 | 33.46 | 37.42 | 24.98 | 22.40 |
| | Homogenous-Small | 10 | 93.79 | 85.66 | 75.23 | 66.87 | 51.90 | 30.61 | 37.40 | 27.16 | 26.20 |

Table 1: Global model accuracy comparison between baselines and their optimized versions suggested by our theory. We observe improved performance on almost all optimized results, especially on subnet-extraction-based methods on high model reduction levels.

data with participation ratio $c = 0.1$ which will include 10 random active clients per communication round. Please see the appendix for other experiment details.

**Data Heterogeneity.** We follow previous works [3, 5] to model non-IID data distribution by limiting the maximum number of labels $L$ as each client is accessing. We consider two levels of data heterogeneity: for MNIST and CIFAR-10 we consider $L = 2$ as high data heterogeneity and $L = 5$ as low data heterogeneity as used in [9]. For CIFAR-100 we consider $L = 20$ as high data heterogeneity and $L = 50$ as low data heterogeneity. This will correspond to an approximate setting of $Dir_K(\alpha)$ with $\alpha = 0.1$ for MNIST, $\alpha = 0.1$ for CIFAR-10, and $\alpha = 0.5$ for CIFAR-100 respectively in Dirichlet-distribution-based data heterogeneity.

**Model Heterogeneity.** In our evaluation, we consider the following client model reduction levels: $\boldsymbol{\beta} = \{1, \frac{3}{4}, \frac{1}{2}, \frac{1}{4}\}$ for MLP and $\boldsymbol{\beta} = \{1, \frac{1}{2}, \frac{1}{4}, \frac{1}{8}\}$ for ResNet, where each fraction represents its model capacity ratio to the largest client model (full model). To generate these client models, for MLP we reduce the number of nodes in each hidden layer, for WResNet we reduce the number of kernels in convolution layers while keeping the nodes in the output layer as the original.

**Baselines and Testcase Notations.** As this experiment is mainly to validate the proposed theory and gather empirical findings, we choose the standard federated learning algorithm, i.e. FedAvg [1], with several different heterogeneous FL model settings. Since this experiment section is to verify the impact of our proposed theory rather than chasing a SOTA accuracy, no further tuning or tricks for training were used to demonstrate the impacts of key factors from the main theorems.

We consider 3 levels of model reduction through pruning and static subnet Extraction: which will reduce the model by only keeping the largest or leading $\beta$ percentile of the parameters per layer. We show 4 homogeneous settings with the full model and the models with 3 levels of model reduction, each with at least one full model so that $\Gamma_{min} > 1$ is achieved. Finally, we consider manually increasing the minimum coverage index and present one possible case denoted as "optimized", by applying local models covering different regions of the global model as illustrated in Fig 1.

Note that even when given a specific model reduction level and the minimum coverage index, there could be infinite combinations of local model reduction solutions; at the same time model reduction will inevitably lead to an increased model reduction noise, by conducting only weights pruning will bring the lowest model reduction noise for a certain model reduction level. How to manage the trade-off between increasing $\Gamma_{min}$ while keeping $\delta^2$ low is non-trivial and will be left for future works on designing effective model reduction policies for heterogeneous FL.

### 6.2 Numerical Results and Further Discussion

We summarize the testing results with one optimized version for comparison in Table 1. We plot the training results of Heterogeneous FL with IID and non-IID data on the MNIST dataset in Figure 2a and Figure 2b, since the model and its reduction are closer to the theoretical setup and its assumptions. We only present training results of medium-level model reduction (where we deploy 4 clients with

fullmodel and 6 clients with $\frac{3}{4}$ models) in the figure at the main paper due to page limit and simplicity. We leave further details and more results in the appendix.

**General Results.** Overall, we observe improved performance on almost all optimized results, especially on subnet-extraction-based methods on high model reduction levels. In most cases, performances will be lower compared to the global model due to model-reduction noise.

**Impact of model-reduction noise.** As our analysis suggests, one key factor affecting convergence is model-reduction noise $\delta^2$. When a model is reduced, inevitably the model-reduction noise $\delta^2$ will affect convergence and model accuracy. Yet, our analysis shows that increasing local epochs or communication rounds cannot mitigate such noise. To minimize the convergence gap in the upperbounds, it is necessary to design model reduction strategies in heterogeneous FL with respect to both model-reduction noise and minimum coverage index, e.g., by considering a joint objective of preserving large parameters while sufficiently covering all parameters.

**Impact of minimum coverage index.** Our theory suggests that for a given model reduction noise, the minimum coverage index $\Gamma_{\min}$ is inversely proportional to the convergence gap as the bound in Theorem 1 indicates. Then for a given model reduction level, a model reduction strategy in heterogeneous FL with a higher minimum coverage index may result in better training performance. Note that existing heterogeneous FL algorithms with pruning often focus on removing the small model parameters that are believed to have an insignificant impact on model performance, while being oblivious to the coverage of parameters in pruned local models, and the model-extraction-based method will only keep the leading subnet. Our analysis in this paper highlights this important design for model reduction strategies in heterogeneous FL that parameter coverages matter.

**More discussions and empirical findings.** For the trade-off between minimum coverage index and model reduction noise, it's nearly impossible to fix one and investigate the impact of the other. In addition, we found: (1) Large models hold more potential to be reduced while maintaining generally acceptable accuracy. (2) Smaller models tend to be affected more by $\delta^2$ while the larger model is more influenced by $\Gamma_{min}$, which suggests that it's more suitable to apply pruning on small networks and apply subnet extraction on large networks.

**Limitations** In this work we consider full device participation, where arbitrary partial participation scenario is not considered. Also, the optimal design of model extraction maintaining a balance between a low $\delta^2$ and a high $\Gamma_{\min}$ is highly non-trivial which would be left for future work.

## 7 Conclusion

In this paper, we present a unifying framework and establish sufficient conditions for FL with dynamic heterogeneous client-dependent local models to converge to a small neighborhood of a stationary point of standard FL. The optimality gap is characterized and depends on model reduction noise and a new concept of minimum coverage index. The result recovers a number of state-of-the-art FL algorithms as special cases. It also provides new insights on designing optimized model reduction strategies in heterogeneous FL, with respect to both minimum coverage index $\Gamma_{\min}$ and model reduction noise $\delta^2$. We empirically demonstrated the correctness of the theory and the design insights. Our work contributes to a deeper theoretical comprehension of heterogeneous FL with adaptive local model reduction and offers valuable insights for the development of new FL algorithms in future research.

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
