| Low Model Reduction | Pruning-Greedy | 6 | 98.18 | 97.60 | 93.15 | 72.66 | 62.49 | 57.17 | 67.41 | 65.67 | 65.06 |
| | Pruning-Optimised | 8 | 98.53 | 98.25 | 95.85 | 76.26 | 66.25 | 59.98 | 67.47 | 66.56 | 67.47 |
| | Static Subnet Subtraction | 6 | 97.62 | 95.12 | 92.33 | 73.20 | 61.25 | 56.09 | 66.60 | 65.24 | 65.79 |
| | Subnet Subtraction - Optimised | 8 | 97.76 | 94.41 | 93.60 | 73.78 | 64.17 | 58.09 | 67.81 | 66.66 | 67.23 |
| | Homogenous-Large | 10 | 97.52 | 96.08 | 93.23 | 69.05 | 63.72 | 57.42 | 66.81 | 65.57 | 63.90 |
| Medium Model Reduction | Pruning-Greedy | 4 | 97.51 | 95.05 | 91.86 | 66.85 | 60.93 | 56.98 | 52.92 | 45.88 | 45.68 |
| | Pruning-Optimised | 8 | 98.39 | 98.02 | 95.48 | 71.43 | 66.94 | 56.93 | 55.32 | 46.81 | 45.74 |
| | Static Subnet Subtraction | 4 | 95.56 | 92.33 | 92.05 | 61.87 | 58.08 | 46.03 | 50.59 | 44.22 | 45.25 |
| | Subnet Subtraction - Optimised | 8 | 97.96 | 94.05 | 93.36 | 63.96 | 62.65 | 47.44 | 52.95 | 46.23 | 46.15 |
| | Homogenous-Medium | 10 | 97.05 | 92.71 | 90.82 | 59.21 | 57.61 | 53.43 | 52.19 | 36.08 | 34.06 |
| High Model Reduction | Pruning-Greedy | 3 | 95.01 | 86.83 | 76.64 | 67.35 | 56.75 | 22.55 | 39.29 | 26.14 | 25.97 |
| | Pruning-Optimised | 5 | 95.32 | 91.98 | 81.66 | 67.74 | 57.33 | 27.97 | 40.78 | 29.63 | 26.63 |
| | Static Subnet Subtraction | 3 | 95.88 | 81.64 | 71.64 | 68.78 | 56.88 | 30.61 | 41.18 | 27.55 | 26.23 |

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

# A  Proof of Theorems

## A.1  Problem summary and notations

We summarize the algorithm in a way that can present the convergence analysis more easily. We use a superscript such as $\theta^{(i)}$, $m_{q,n}^{(i)}$, and $\nabla F^{(i)}$ to denote the sub-vector of parameter, mask, and gradient corresponding to region $i$. For the proof purpose and with slight abuse of notations, we denote all modeling parameters contained in the same set of local models as a parameter region $i$ (Ultimately we can regard each modeling parameter as a separate region). In each round $q$, parameters in each region $i$ is contained in and only in a set of local models denoted by $\mathcal{N}_q^{(i)}$, implying that $m_{q,n}^{(i)} = \mathbf{1}$ for $n \in \mathcal{N}_q^{(i)}$ and $m_{q,n}^{(i)} = \mathbf{0}$ otherwise, for all the parameters in the region. We define $\Gamma^* = \min_{q,i} \mathcal{N}_q^{(i)}$ as the minimum coverage index, since it denotes the minimum number of local models that contain any parameters in $\theta_q$. With slight abuse of notations, we use $\nabla F_n(\theta$ and $\nabla F_n(\theta, \xi)$ to denote the gradient and stochastic gradient, respectively.

---

**Algorithm 1:** The unifying heterogenous FL framework.

---

**Input:** Local data $D_i^k$ on $N$ clients, reduction policy $\mathbb{P}$.
**Executes:**
Initialize $\theta_0$
**for** *round $q = 1, 2, \ldots, Q$* **do**
    **for** *local workers $n = 1, 2, \ldots, N$ (In parallel)* **do**
        Generate model reduction mask $m_{q,n} = \mathbb{P}(\theta_q, n)$
        Generate local models $\theta_{q,n,0} = \theta_q \odot m_{q,n}$
        // Update local models:
        **for** *epoch $t = 1, 2, \ldots, T$* **do**
            $\theta_{q,n,t} = \theta_{q,n,t-1} - \gamma \nabla F_n(\theta_{q,n,t-1}, \xi_{n,t-1}) \odot m_{q,n}$
        **end**
    **end**
    // Update global model:
    **for** *region $i = 1, 2, \ldots, K$* **do**
        Find $\mathcal{N}_q^{(i)} = \{n : m_{q,n}^{(i)} = \mathbf{1}\}$
        Update $\theta_{q+1}^{(i)} = \frac{1}{|\mathcal{N}_q^{(i)}|} \sum_{n \in \mathcal{N}_q^{(i)}} \theta_{q,n,T}^{(i)}$
    **end**
**end**
Output $\theta_Q$

---

## A.2  Nomenclature

We present Table 1 to better summarize and explain the notations used. A more detailed explanation of each term is available when they are first introduced in the main paper.

## A.3  Assumptions

**Assumption 1.** *(Smoothness). Cost functions $F_1, \ldots, F_N$ are all L-smooth: $\forall \theta, \phi \in \mathcal{R}^d$ and any $n$, we assume that there exists $L > 0$:*

$$\|\nabla F_n(\theta) - \nabla F_n(\phi)\| \leq L \|\theta - \phi\|. \tag{1}$$

**Assumption 2.** *(model reduction noise). We assume that for some $\delta^2 \in [0, 1)$ and any $q, n, t$, the model reduction noise is bounded by*

$$\|\theta_{q,n,t} - \theta_{q,n,t} \odot m_{q,n}\|^2 \leq \delta^2 \|\theta_{q,n,t}\|^2. \tag{2}$$

**Assumption 3.** *(Bounded Gradient). The expected squared norm of stochastic gradients is bounded uniformly, i.e., for constant $G > 0$ and any $n, q, t$:*

$$E \|\nabla F_n(\theta_{q,n,t}, x_{q,n,t})\|^2 \leq G. \tag{3}$$

| Notation | Explanation |
|---|---|
| $q, Q$ | Current and Total communication round |
| $n, N$ | Local client, total client number |
| $i, K$ | Region (or set of parameters), total region |
| $\theta_q$ | Global model at q-th round |
| $m_{q,n}$ | Model reduction mask |
| $\mathcal{N}_q^{(i)}$ | Parameter set, whose local models contain the $i$th modeling parameter or i-th region in round $q$ |
| $\mathbb{P}$ | Model reduction method |
| $\nabla F_n(\theta)$ | Local stochastic gradient |
| $\xi$ | Sampled training data |
| $D_n$ | Data distribution |
| $\mathcal{N}^{(i)}$ | Number of local models that containing the $i$th parameter/region |
| $\delta^2$ | Model reduction ratio |
| $\sigma^2$ | Gradient variance bound |
| $G$ | Stochastic gradients bound |
| $\Gamma_{min}$ | Minimum covering index |

Table 1

**Assumption 4.** *(Gradient Noise for IID data). Under IID data distribution, for any $q, n, t$, we assume that*

$$\mathbb{E}[\nabla F_n(\theta_{q,n,t}, \xi_{n,t})] = \nabla F(\theta_{q,n,t}) \tag{4}$$

$$\mathbb{E}\|\nabla F_n(\theta_{q,n,t}, \xi_{n,t}) - \nabla F(\theta_{q,n,t})\|^2 \leq \sigma^2 \tag{5}$$

*where $\sigma^2 > 0$ is a constant and $\xi_{n,t}$) are independent samples for different $n, t$.*

**Assumption 5.** *(Gradient Noise for non-IID data). Under non-IID data distribution, we assume that for constant $\sigma^2 > 0$ and any $q, n, t$:*

$$\mathbb{E}\left[\frac{1}{|\mathcal{N}_q^{(i)}|} \sum_{n \in \mathcal{N}_q^{(i)}} \nabla F_n^{(i)}(\theta_{q,n,t}, \xi_{n,t})\right] = \nabla F^{(i)}(\theta_{q,n,t}) \tag{6}$$

$$\mathbb{E}\left\|\frac{1}{|\mathcal{N}_q^{(i)}|} \sum_{n \in \mathcal{N}_q^{(i)}} \nabla F_n^{(i)}(\theta_{q,n,t}, \xi_{n,t}) - \nabla F^{(i)}(\theta_{q,n,t})\right\|^2 \leq \sigma^2. \tag{7}$$

### A.4 Convergence Analysis

We now analyze the convergence of heterogeneous FL under adaptive online model pruning with respect to any pruning policy $\mathbb{P}(\theta_q, n)$ (and the resulting mask $m_{q,n}$) and prove the main theorems in this paper. We need to overcome a number of challenges as follows:

- We will begin the proof by analyzing the change of loss function in one round as the model goes from $\theta_q$ to $\theta_{q+1}$, i.e., $F(\theta_{q+1}) - F(\theta_1)$. It includes three major steps: pruning to obtain heterogeneous local models $\theta_{q,n,0} = \theta_q \odot m_{q,n}$, training local models in a distributed fashion to update $\theta_{q,n,t}$, and parameter aggregation to update the global model $\theta_{q+1}$.
- Due to the use of heterogeneous local models whose masks $m_{q,n}$ both vary over rounds and change for different workers, we first characterize the difference between local model $\theta_{q,n,t}$ at any epoch $t$ and global model $\theta_q$ at the beginning of the current round. It is easy to see that this can be factorized into two parts: model reduction noise $\|\theta_{q,n,0} - \theta_q\|^2$ and local training $\|\theta_{q,n,t} - \theta_{q,n,0}\|^2$, which will be analyzed in Lemma 1.
- We characterize the impact of heterogeneous local models on global parameter update. Specifically, we use an ideal local gradient $\nabla F_n(\theta_q)$ as a reference point and quantify the different between aggregated local gradients and the ideal gradient. This will be presented in Lemma 2. We also quantify the norm difference between a gradient and a stochastic gradient (with respect to the global update step) using the gradient noise assumptions, in Lemma 3.
- Since IID and non-IID data distributions in our model differ in the gradient noise assumption (i.e., Assumption 4 and Assumption 5), we present a unified proof for both cases. We will explicitly state IID and non-IID data distributions only if the two cases require different treatment (when the gradient noise assumptions are needed). Otherwise, the derivations and proofs are identical for both cases.

We will begin by proving a number of lemmas and then use them for convergence analysis.

**Lemma 1.** *Under Assumption 2 and Assumption 3, for any q, we have:*

$$\sum_{t=1}^{T}\sum_{n=1}^{N}\mathbb{E}\|\theta_{q,n,t-1}-\theta_q\|^2 \leq \frac{2\gamma^2 T^3 NG}{3} + 2\delta^2 NT \cdot \mathbb{E}\|\theta_q\|^2. \tag{8}$$

*Proof.* We note that $\theta_q$ is the global model at the beginning of current round. We split the difference $\theta_{q,n,t-1}-\theta_q$ into two parts: changes due to local model training $\theta_{q,n,t-1}-\theta_{q,n,0}$ and changes due to pruning $\theta_{q,n,0}-\theta_q$. That is

$$\sum_{t=1}^{T}\sum_{n=1}^{N}\mathbb{E}\|\theta_{q,n,t-1}-\theta_q\|^2$$

$$=\sum_{t=1}^{T}\sum_{n=1}^{N}\mathbb{E}\| \left(\theta_{q,n,t-1}-\theta_{q,n,0}\right) + \left(\theta_{q,n,0}-\theta_q\right)\|^2$$

$$\leq\sum_{t=1}^{T}\sum_{n=1}^{N}2\mathbb{E}\|\theta_{q,n,t-1}-\theta_q\|^2 + \sum_{t=1}^{T}\sum_{n=1}^{N}2\mathbb{E}\|\theta_{q,n,t-1}-\theta_q\|^2 \tag{9}$$

where we used the fact that $\|\sum_{i=1}^{s}a_i\|^2 \leq s\sum_{i=1}^{s}\|a_i\|^2$ in the last step.

For the first term in Eq.(9), we notice that $\theta_{q,n,t-1}$ is obtained from $\theta_{q,n,0}$ through $t-1$ epochs of local model updates on worker $n$. Using the local gradient updates from the algorithm, it is easy to see:

$$\sum_{t=1}^{T}\sum_{n=1}^{N}\mathbb{E}\|\theta_{q,n,t-1}-\theta_{q,n,0}\|^2$$

$$=\sum_{t=1}^{T}\sum_{n=1}^{N}\mathbb{E}\left\|\sum_{j=1}^{t-1}-\gamma\nabla F_n\left(\theta_{q,n,j-1};\xi_{n,j-1}\right)\odot m_{q,n}\right\|^2$$

$$\leq\sum_{t=1}^{T}\sum_{n=1}^{N}(t-1)\sum_{j=1}^{t-1}\mathbb{E}\left\|-\gamma\nabla F_n\left(\theta_{q,n,j-1};\xi_{n,j-1}\right)\odot m_{q,n}\right\|^2$$

$$\leq\sum_{t=1}^{T}\sum_{n=1}^{N}(t-1)\sum_{j=1}^{t-1}\gamma^2 G$$

$$\leq\gamma^2 NG\sum_{t=1}^{T}(t-1)^2$$

$$\leq\frac{\gamma^2 T^3 NG}{3}, \tag{10}$$

where we use the fact that $\|\sum_{i=1}^{s}a_i\|^2 \leq s\sum_{i=1}^{s}\|a_i\|^2$ in step 2 above, and the fact that $m_{q,n}$ is a binary mask in step 3 above together with Assumption 3 for bounded gradient.

For the second term in Eq.(9), the difference is resulted by model pruning using mask $m_{n,q}$ of work $n$ in round $q$. We have

$$\sum_{t=1}^{T}\sum_{n=1}^{N}\mathbb{E}\|\theta_{q,n,0}-\theta_q\|^2 \quad =\sum_{t=1}^{T}\sum_{n=1}^{N}\mathbb{E}\|\theta_q\odot m_{n,q}-\theta_q\|^2$$

$$\leq\sum_{t=1}^{T}\sum_{n=1}^{N}\delta^2\mathbb{E}\|\theta_q\|^2$$

$$=\delta^2 NT\cdot\mathbb{E}\|\theta_q\|^2, \tag{11}$$

where we used the fact that $\theta_{q,n,0} = \theta_q \odot m_{n,q}$ in step 1 above, and Assumption 2 in step 2 above. Plugging Eq.(10) and Eq.(11) into Eq.(9), we obtain the desired result. $\qquad\square$

**Lemma 2.** *Under Assumptions 1-3, for any q, we have:*

$$\sum_{i=1}^{K} \mathbb{E} \left\| \frac{1}{\Gamma_q^{(i)}T} \sum_{t=1}^{T} \sum_{n \in \mathcal{N}_q^{(i)}} \left[ \nabla F_n^{(i)}(\theta_{q,n,t-1}) - \nabla F_n^{(i)}(\theta_q) \right] \right\|^2$$
$$\leq \frac{L^2 \gamma^2 TNG}{\Gamma^*} + \frac{L^2 \delta^2 N}{\Gamma^*} \mathbb{E}\|\theta_q\|^2. \tag{12}$$

*Proof.* Recall that $\Gamma_q^{(i)} = |\mathcal{N}_q^{(i)}|$ is the number of local models containing parameters of region $i$ in round $q$. The left-hand-side of Eq.(12) denotes the difference between an average gradient of heterogeneous models (through aggregation and over time) and an ideal gradient. The summation over $i$ adds up such difference over all regions $i = 1, \ldots, K$, because the average gradient takes a different form in different regions.

From the inequality $\| \sum_{i=1}^{s} a_i \|^2 \leq s \sum_{i=1}^{s} \|a_i\|^2$, we obtain $\|\frac{1}{s} \sum_{i=1}^{s} a_i\|^2 \leq \frac{1}{s} \sum_{i=1}^{s} \|a_i\|^2$. We use this inequality on the left-hand-side of Eq.(12) to get:

$$\sum_{i=1}^{K} \mathbb{E} \left\| \frac{1}{\Gamma_q^{(i)}T} \sum_{t=1}^{T} \sum_{n \in \mathcal{N}_q^{(i)}} \left[ \nabla F_n^{(i)}(\theta_{q,n,t-1}) - \nabla F_n^{(i)}(\theta_q) \right] \right\|^2$$
$$\leq \sum_{i=1}^{K} \frac{1}{\Gamma_q^{(i)}T} \sum_{t=1}^{T} \sum_{n \in \mathcal{N}_q^{(i)}} \mathbb{E} \left\| \nabla F_n^{(i)}(\theta_{q,n,t-1}) - \nabla F_n^{(i)}(\theta_q) \right\|^2$$
$$\leq \frac{1}{T\Gamma^*} \sum_{t=1}^{T} \sum_{n=1}^{N} \sum_{i=1}^{K} \mathbb{E} \left\| \nabla F_n^{(i)}(\theta_{q,n,t-1}) - \nabla F_n^{(i)}(\theta_q) \right\|^2$$
$$= \frac{1}{T\Gamma^*} \sum_{t=1}^{T} \sum_{n=1}^{N} \mathbb{E} \left\| \nabla F_n(\theta_{q,n,t-1}) - \nabla F_n(\theta_q) \right\|^2$$
$$\leq \frac{1}{T\Gamma^*} \sum_{t=1}^{T} \sum_{n=1}^{N} L^2 \mathbb{E} \left\| \theta_{q,n,t-1} - \theta_q \right\|^2, \tag{13}$$

where we relax the inequality by choosing the smallest $\Gamma^* = \min_{q,i} \Gamma_q^{(i)}$ and changing the summation over $n$ to all workers in the second step. In the third step, we use the fact that $L_2$ gradient norm of a vector is equal to the sum of norm of all sub-vectors (i.e., regions $i = 1, \ldots, K$). This allows us to consider $\nabla F_n$ instead of its sub-vectors on different regions.

Finally, the last step is directly from L-smoothness in Assumption 1. Under Assumptions 2-3, we notice that the last step of Eq.(13) is further bounded by Lemma 1, which yields the desired result of this lemma after re-arranging the terms. $\qquad\square$

**Lemma 3.** *For IID data distribution under Assumptions 4, for any q, we have:*

$$\sum_{i=1}^{K} \mathbb{E} \left\| \frac{1}{\Gamma_q^{(i)}T} \sum_{t=1}^{T} \sum_{n \in \mathcal{N}_q^{(i)}} \left[ \nabla F_n^{(i)}(\theta_{q,n,t-1}, \xi_{n,t-1}) - \nabla F^{(i)}(\theta_{q,n,t-1}) \right] \right\|^2 \leq \frac{N\sigma^2}{T(\Gamma^*)^2}.$$

*For non-IID data distribution under Assumption 5, for any q, we have:*

$$\sum_{i=1}^{K} \mathbb{E} \left\| \frac{1}{\Gamma_q^{(i)}T} \sum_{t=1}^{T} \sum_{n \in \mathcal{N}_q^{(i)}} \left[ \nabla F_n^{(i)}(\theta_{q,n,t-1}, \xi_{n,t-1}) - \nabla F^{(i)}(\theta_{q,n,t-1}) \right] \right\|^2 \leq \frac{K\sigma^2}{T}.$$

*Proof.* This lemma quantifies the square norm of the difference between gradient and stochastic gradient in the global parameter update. We present results for both IID and non-IID cases in this lemma under Assumption 4 and Assumption 5, respectively.

We first consider IID data distributions. Since all the samples $\xi_{n,t-1}$ are independent from each other for different $n$ and $t-1$, the difference between gradient and stochastic gradient, i.e., $\nabla F_n^{(i)}(\theta_{q,n,t-1},\xi_{n,t-1}) - \nabla F_n^{(i)}(\theta_{q,n,t-1})$, are independent gradient noise. Due to Assumption 4, these gradient noise has zero mean. Using the fact that $\mathbb{E}\|\sum_i \mathbf{x}_i\|^2 = \sum_i \mathbb{E}\|\mathbf{x}_i^2\|$ for zero-mean and independent $\mathbf{x}_i$'s, we get:

$$\sum_{i=1}^{K}\mathbb{E}\left\|\frac{1}{\Gamma_q^{(i)}T}\sum_{t=1}^{T}\sum_{n\in\mathcal{N}_q^{(i)}}\left[\nabla F_n^{(i)}(\theta_{q,n,t-1},\xi_{n,t-1})-\nabla F_n^{(i)}(\theta_{q,n,t-1})\right]\right\|^2$$

$$\leq \sum_{i=1}^{K}\frac{1}{(\Gamma_q^{(i)}T)^2}\sum_{t=1}^{T}\sum_{n\in\mathcal{N}_q^{(i)}}\mathbb{E}\left\|\nabla F_n^{(i)}(\theta_{q,n,t-1},\xi_{n,t-1})-\nabla F_n^{(i)}(\theta_{q,n,t-1})\right\|^2$$

$$\leq \frac{1}{(T\Gamma^*)^2}\sum_{i=1}^{K}\sum_{t=1}^{T}\sum_{n=1}^{N}\mathbb{E}\left\|\nabla F_n^{(i)}(\theta_{q,n,t-1},\xi_{n,t-1})-\nabla F_n^{(i)}(\theta_{q,n,t-1})\right\|^2$$

$$= \frac{1}{(T\Gamma^*)^2}\sum_{t=1}^{T}\sum_{n=1}^{N}\mathbb{E}\left\|\nabla F_n(\theta_{q,n,t-1},\xi_{n,t-1})-\nabla F_n(\theta_{q,n,t-1})\right\|^2$$

$$\leq \frac{1}{(T\Gamma^*)^2}\cdot TN\sigma^2 \tag{14}$$

where we used the property of zero-mean and independent gradient noise in the first step above, relax the inequality by choosing the smallest $\Gamma^* = \min_{q,i}\Gamma_q^{(i)}$ and changing the summation over $n$ to all workers in the second step. In the third step, we use the fact that $L_2$ gradient norm of a vector is equal to the sum of norm of all sub-vectors (i.e., regions $i = 1,\ldots,K$). This allows us to consider $\nabla F_n$ instead of its sub-vectors on different regions. Finally, we apply Assumption 4 to bound the gradient noise and obtain the desired result.

For non-IID data distributions under Assumption 4 (instead of Assumption 5), we notice that $\mathbb{E}\left[\frac{1}{|\mathcal{N}_q^{(i)}|}\sum_{n\in\mathcal{N}_q^{(i)}}\nabla F_n^{(i)}(\theta_{q,n,t-1},\xi_{n,t-1})\right] = \nabla F^{(i)}(\theta_{q,n,t-1})$ is an unbiased estimate for any epoch $t$, with bounded gradient noise. Again, due to independent samples $\xi_{n,t-1}$, we have:

$$\sum_{i=1}^{K}\mathbb{E}\left\|\frac{1}{\Gamma_q^{(i)}T}\sum_{t=1}^{T}\sum_{n\in\mathcal{N}_q^{(i)}}\left[\nabla F_n^{(i)}(\theta_{q,n,t-1},\xi_{n,t-1})-\nabla F_n^{(i)}(\theta_{q,n,t-1})\right]\right\|^2$$

$$\leq \frac{1}{T^2}\sum_{i=1}^{K}\sum_{t=1}^{T}\mathbb{E}\left\|\frac{1}{\Gamma_q^{(i)}}\sum_{n\in\mathcal{N}_q^{(i)}}\nabla F_n^{(i)}(\theta_{q,n,t-1},\xi_{n,t-1})-\nabla F_n^{(i)}(\theta_{q,n,t-1})\right\|^2$$

$$\leq \frac{1}{T^2}\sum_{i=1}^{K}\sum_{t=1}^{T}\sigma^2$$

$$= \frac{K\sigma^2}{T}, \tag{15}$$

where we use the property of zero-mean and independent gradient noise in the first step above, used the fact that the norm of a sub-vector (in the region $i$) is bounded by that of the entire vector in the second step above, as well as Assumption 5. This completes the proof of this lemma. $\qquad\square$

**Proof of the main result**. Now we are ready to present the main proof. We begin with the L-smoothness property in Assumption 1, which implies

$$F(\theta_{q+1}) - F(\theta_q) \leq \langle \nabla F(\theta_q),\, \theta_{q+1} - \theta_q\rangle + \frac{L}{2}\|\theta_{q+1} - \theta_q\|^2. \tag{16}$$

We take expectations on both sides of the inequality and get:

$$\mathbb{E}[F(\theta_{q+1})] - \mathbb{E}[]F(\theta_q)] \le \mathbb{E}\left\langle \nabla F(\theta_q),\ \theta_{q+1} - \theta_q \right\rangle + \frac{L}{2}\mathbb{E}\left\| \theta_{q+1} - \theta_q \right\|^2. \tag{17}$$

In the following, we bound the two terms on the right-hand side above and finally combine the results to complete the proof.

**Upperbound for** $\mathbb{E}\left\langle \nabla F(\theta_q),\ \theta_{q+1} - \theta_q \right\rangle$. We notice that the inner product can be broken down and reformulated as the sum of inner products over all regions $i = 1, \dots, K$. This is necessary because the global parameter update is different for different regions. More precisely, for any region $i$, we have:

$$
\begin{aligned}
\theta_{q+1}^{(i)} - \theta_q^{(i)} &= \left( \frac{1}{\Gamma_q^{(i)}} \sum_{n \in \mathcal{N}_q^{(i)}} \theta_{q,n,T}^{(i)} \right) - \theta_q^{(i)} \\
&= \frac{1}{\Gamma_q^{(i)}} \sum_{n \in \mathcal{N}_q^{(i)}} \left[ \theta_{q,n,0}^{(i)} - \sum_{t=1}^{T} \gamma \nabla F_n^{(i)}(\theta_{q,n,t-1}, \xi_{n,t-1}) \cdot m_{n,q}^{(i)} \right] - \theta_q^{(i)} \\
&= -\frac{1}{\Gamma_q^{(i)}} \sum_{n \in \mathcal{N}_q^{(i)}} \sum_{t=1}^{T} \gamma \nabla F_n^{(i)}(\theta_{q,n,t-1}, \xi_{n,t-1}) \cdot m_{n,q}^{(i)} + \theta_q^{(i)} \cdot m_{n,q}^{(i)} - \theta_q^{(i)} \\
&= -\frac{1}{\Gamma_q^{(i)}} \sum_{n \in \mathcal{N}_q^{(i)}} \sum_{t=1}^{T} \gamma \nabla F_n^{(i)}(\theta_{q,n,t-1}, \xi_{n,t-1}), \tag{18}
\end{aligned}
$$

where global parameter updated is used in the first step, local parameter update is used in the second step, and the third step follows from the fact that for any worker $n \in \mathcal{N}_q^{(i)}$ participating in the global update of $\theta_q^{(i)}$ contain the model parameters of region $i$, i.e., $m_{q,n}^{(i)} = \mathbf{1}$. We also use $\theta_{q,n,0}^{(i)} = \theta_q^{(i)} \cdot m_{n,q}^{(i)}$ in the third step above because of to pruning.

Next we analyze $\mathbb{E}\left\langle \nabla F(\theta_q),\ \theta_{q+1} - \theta_q \right\rangle$ by considering a sum of inner products over $K$ regions. We have

$$
\begin{aligned}
&\mathbb{E}\left\langle \nabla F(\theta_q),\ \theta_{q+1} - \theta_q \right\rangle \\
&= \sum_{i=1}^{K} \mathbb{E}\left\langle \nabla F^{(i)}(\theta_q),\ \theta_{q+1}^{(i)} - \theta_q^{(i)} \right\rangle \\
&= \sum_{i=1}^{K} \mathbb{E}\left\langle \nabla F^{(i)}(\theta_q),\ -\frac{1}{\Gamma_q^{(i)}} \sum_{n \in \mathcal{N}_q^{(i)}} \sum_{t=1}^{T} \gamma \nabla F_n^{(i)}(\theta_{q,n,t-1}, \xi_{n,t-1}) \right\rangle \\
&= \sum_{i=1}^{K} \mathbb{E}\left\langle \nabla F^{(i)}(\theta_q),\ -\frac{1}{\Gamma_q^{(i)}} \sum_{n \in \mathcal{N}_q^{(i)}} \sum_{t=1}^{T} \gamma \mathbb{E}\left[ \nabla F_n^{(i)}(\theta_{q,n,t-1}, \xi_{n,t-1}) | \theta_q \right] \right\rangle \\
&= \sum_{i=1}^{K} \mathbb{E}\left\langle \nabla F^{(i)}(\theta_q),\ -\frac{1}{\Gamma_q^{(i)}} \sum_{n \in \mathcal{N}_q^{(i)}} \sum_{t=1}^{T} \gamma \nabla F_n^{(i)}(\theta_{q,n,t-1}) \right\rangle \\
&= -\sum_{i=1}^{K} \mathbb{E}\left\langle \nabla F^{(i)}(\theta_q),\ \gamma T \nabla F^{(i)}(\theta_q) \right\rangle \tag{19} \\
&\quad -\sum_{i=1}^{K} \mathbb{E}\left\langle \nabla F^{(i)}(\theta_q),\ \frac{1}{\Gamma_q^{(i)}} \sum_{n \in \mathcal{N}_q^{(i)}} \sum_{t=1}^{T} \gamma \left[ \nabla F_n^{(i)}(\theta_{q,n,t-1}) - \nabla F^{(i)}(\theta_q) \right] \right\rangle
\end{aligned}
$$

where we use the first step to reformulate the inner product as a sum, the second step follows from Eq.(18), the third step employs a conditional expectation over the random samples with respect to $\theta_q$, and the last step splits the result into two parts with respect to a reference point $\gamma T \nabla F^{(i)}(\theta_q)$.

For the first term on the right-hand side of Eq.(19), it is easy to see that

$$-\sum_{i=1}^{K} \mathbb{E} \left\langle \nabla F^{(i)}(\theta_q),\, \gamma T \nabla F^{(i)}(\theta_q) \right\rangle = -\gamma T \sum_{i=1}^{K} \left\| \nabla F^{(i)}(\theta_q) \right\|^2$$
$$= -\gamma T \mathbb{E} \left\| \nabla F(\theta_q) \right\|^2, \qquad (20)$$

where we add up the norm over $K$ regions in the last step. For the second term on the right-hand-side of Eq.(19), we use the inequality $< a, b > \le \frac{1}{2}\|a\|^2 + \frac{1}{2}\|b\|^2$ for any vectors $a, b$. Applying this inequality to the second term, we have

$$-\sum_{i=1}^{K} \mathbb{E} \left\langle \nabla F^{(i)}(\theta_q),\, \frac{1}{\Gamma_q^{(i)}} \sum_{n \in \mathcal{N}_q^{(i)}} \sum_{t=1}^{T} \gamma \left[ \nabla F_n^{(i)}(\theta_{q,n,t-1}) - \nabla F^{(i)}(\theta_q) \right] \right\rangle$$

$$= -\sum_{i=1}^{K} T\gamma \cdot \mathbb{E} \left\langle \nabla F^{(i)}(\theta_q),\, \frac{1}{T\Gamma_q^{(i)}} \sum_{n \in \mathcal{N}_q^{(i)}} \sum_{t=1}^{T} \left[ \nabla F_n^{(i)}(\theta_{q,n,t-1}) - \nabla F^{(i)}(\theta_q) \right] \right\rangle$$

$$\le \frac{T\gamma}{2} \sum_{i=1}^{K} \mathbb{E} \left\| \nabla F^{(i)}(\theta_q) \right\|^2 + \frac{T\gamma}{2} \sum_{i=1}^{K} \mathbb{E} \left\| \frac{1}{T\Gamma_q^{(i)}} \sum_{n \in \mathcal{N}_q^{(i)}} \sum_{t=1}^{T} \left[ \nabla F_n^{(i)}(\theta_{q,n,t-1}) - \nabla F^{(i)}(\theta_q) \right] \right\|$$

$$= \frac{T\gamma}{2} \mathbb{E} \left\| \nabla F(\theta_q) \right\|^2 + \frac{T\gamma}{2} \left( \frac{L^2 \gamma^2 TNG}{\Gamma^*} + \frac{L^2 \delta^2 N}{\Gamma^*} \mathbb{E}\|\theta_q\|^2 \right) \qquad (21)$$

where the second step uses the inequality and the third step follows directly from Lemma 2. Plugging Eq.(20) and Eq.(21) results into Eq.(19), we obtain the desired upperbound:

$$\mathbb{E} \left\langle \nabla F(\theta_q),\, \theta_{q+1} - \theta_q \right\rangle \le -\frac{T\gamma}{2} \mathbb{E} \left\| \nabla F(\theta_q) \right\|^2 + \frac{T\gamma}{2} \left( \frac{L^2 \gamma^2 TNG}{\Gamma^*} + \frac{L^2 \delta^2 N}{\Gamma^*} \mathbb{E}\|\theta_q\|^2 \right). \qquad (22)$$

**Upperbound for $\frac{L}{2} \mathbb{E} \|\theta_{q+1} - \theta_q\|^2$.** We use the again result in Eq.(18) and apply it to $\theta_{q+1} - \theta_q$, which gives:

$$\frac{L}{2} \mathbb{E} \|\theta_{q+1} - \theta_q\|^2$$

$$= \frac{L}{2} \mathbb{E} \left\| \frac{1}{\Gamma_q^{(i)}} \sum_{n \in \mathcal{N}_q^{(i)}} \sum_{t=1}^{T} \gamma \nabla F_n^{(i)}(\theta_{q,n,t-1}, \xi_{n,t-1}) \right\|^2$$

$$\le \frac{3L}{2} \mathbb{E} \left\| \frac{1}{\Gamma_q^{(i)}} \sum_{n \in \mathcal{N}_q^{(i)}} \sum_{t=1}^{T} \gamma \left[ \nabla F_n^{(i)}(\theta_{q,n,t-1}, \xi_{n,t-1}) - \nabla F_n^{(i)}(\theta_{q,n,t-1}) \right] \right\|^2$$

$$+ \frac{3L}{2} \mathbb{E} \left\| \frac{1}{\Gamma_q^{(i)}} \sum_{n \in \mathcal{N}_q^{(i)}} \sum_{t=1}^{T} \gamma \left[ \nabla F_n^{(i)}(\theta_{q,n,t-1}) - \nabla F_n^{(i)}(\theta_q) \right] \right\|^2$$

$$+ \frac{3L}{2} \mathbb{E} \left\| \frac{1}{\Gamma_q^{(i)}} \sum_{n \in \mathcal{N}_q^{(i)}} \sum_{t=1}^{T} \gamma \nabla F_n^{(i)}(\theta_q) \right\|^2, \qquad (23)$$

where in the second step, we use the inequality $\left\| \sum_{i=1}^{s} a_i \right\|^2 \le s \sum_{i=1}^{s} \|a_i\|^2$ and split stochastic gradient $[\nabla F_n^{(i)}(\theta_{q,n,t-1}, \xi_{n,t-1})]$ into $s = 3$ parts, i.e., $[\nabla F_n^{(i)}(\theta_{q,n,t-1}, \xi_{n,t-1}) - \nabla F_n^{(i)}(\theta_{q,n,t-1})]$, $[F_n^{(i)}(\theta_{q,n,t-1}) - F_n^{(i)}(\theta_q)]$, and $[F_n^{(i)}(\theta_q)]$.

Next, we notice that the third term on the right-hand side of Eq.(23) can be simplified, because (i) for IID data distribution, the cost function of each worker $n$ is the same as the global cost function,

i.e., $\nabla F_n(\theta_q) = \nabla F(\theta_q)$, and (ii) for non-IID data distribution, the gradient noise assumption (Assumption 5) implies that $\frac{1}{\Gamma_q^{(i)}} \sum_{n \in \mathcal{N}_q^{(i)}} \nabla F_n(\theta_q) = F(\theta_q)$. Thus in both cases, we have:

$$\frac{3L}{2} \mathbb{E} \left\| \frac{1}{\Gamma_q^{(i)}} \sum_{n \in \mathcal{N}_q^{(i)}} \sum_{t=1}^{T} \gamma \nabla F_n^{(i)}(\theta_q) \right\|^2 \leq \frac{3LT^2\gamma^2}{2} \sum_{i=1}^{K} \mathbb{E} \|\nabla F^{(i)}(\theta_q)\|^2$$

$$= \frac{3LT^2\gamma^2}{2} \mathbb{E} \|\nabla F(\theta_q)\|^2, \tag{24}$$

where we again used the sum of the norm of $K$ regions in the last step.

Now we notice that the first and second terms of Eq.(23) have been bounded by Lemma 2 and Lemma 3, except for constants $\gamma$ and $1/T$. Applying these results directly and also plugging in Eq.(24) into Eq.(23), we obtain the desired upperbound:

$$\frac{L}{2} \mathbb{E} \|\theta_{q+1} - \theta_q\|^2 \leq \frac{3LTN\gamma^2\sigma^2}{2(\Gamma^*)^2} \text{ (for IID) or } \frac{3LTK\gamma^2\sigma^2}{2} \text{ (for non-IID)}$$

$$+ \frac{3L^3\gamma^4 T^3 NG}{2\Gamma^*} + \frac{3L^3 T^2\gamma^2\delta^2 N}{2\Gamma^*} \mathbb{E} \|\theta_q\|^2$$

$$+ \frac{3LT^2\gamma^2}{2} \mathbb{E} \|\nabla F_n(\theta_q)\|^2. \tag{25}$$

$$\theta_{q,n,t} = \theta_{q,n,t-1} - \gamma \nabla F_n(\theta_{q,n,t-1}; \xi_{n,t-1}) \tag{26}$$

**Combining the two Upperbounds.** Finally, we will apply the upperbound for $\mathbb{E} \langle \nabla F(\theta_q), \theta_{q+1} - \theta_q \rangle$ in Eq.(22) as well as the upperbound for $\frac{L}{2} \mathbb{E} \|\theta_{q+1} - \theta_q\|^2$ in Eq.(25), and plug them into Eq.(17). First we take the sum over $q = 1, \ldots, Q$ on both sides of Eq.(17), which becomes:

$$\mathbb{E}[F(\theta_{Q+1})] - \mathbb{E}[F(\theta_0)]$$

$$= \sum_{q=1}^{Q} \mathbb{E}[F(\theta_{q+1})] - \sum_{q=1}^{Q} \mathbb{E}[F(\theta_q)]$$

$$\leq \sum_{q=1}^{Q} \mathbb{E} \langle \nabla F(\theta_q), \theta_{q+1} - \theta_q \rangle + \sum_{q=1}^{Q} \frac{L}{2} \mathbb{E} \|\theta_{q+1} - \theta_q\|^2. \tag{27}$$

Now plugging in the two upperbounds and re-arranging the terms, for IID data distribution, we derive:

$$\mathbb{E}[F(\theta_{Q+1})] - \mathbb{E}[F(\theta_0)]$$

$$\leq -\frac{T\gamma}{2}(1 - 3LT\gamma) \sum_{q=1}^{Q} \mathbb{E} \|\nabla F(\theta_q)\|^2$$

$$+ \frac{\gamma TQ}{2} \left( \frac{TL^2\gamma^2 NG}{\Gamma^*} + \frac{3LN\gamma\sigma^2}{(\Gamma^*)^2} + \frac{3L^3\gamma^3 T^3 NG}{\Gamma^*} \right)$$

$$+ \frac{T\gamma}{2} \left( \frac{L^2\delta^2 N}{\Gamma^*} + \frac{3L^3 T\gamma\delta^2 N}{\Gamma^*} \right) \sum_{q=1}^{Q} \mathbb{E} \|\theta_q\|^2. \tag{28}$$

We choose learning rate $\gamma \leq 1/(6LT)$ and use the fact that $\mathbb{E}[F(\theta_{Q+1})]$ is non-negative. The inequality above becomes:

$$\frac{T\gamma}{4} \sum_{q=1}^{Q} \mathbb{E} \|\nabla F(\theta_q)\|^2 \leq \mathbb{E}[F(\theta_0)] + \frac{T\gamma Q}{2} \left( \frac{3LN\gamma\sigma^2}{(\Gamma^*)^2} + \frac{3L^2\gamma^2 TNG}{2\Gamma^*} \right)$$

$$+ \frac{T\gamma}{2} \left( \frac{3L^2\delta^2 N}{2\Gamma^*} \right) \sum_{q=1}^{Q} \mathbb{E} \|\theta_q\|^2. \tag{29}$$

Dividing both sides above by $4/(QT\gamma)$ and choosing $\gamma \le 1/T\sqrt{Q}$, we have:

$$\frac{1}{Q}\sum_{q=1}^{Q}\mathbb{E}\|\nabla F(\theta_q)\|^2 \quad \le \frac{4\mathbb{E}[F(\theta_0)]}{\sqrt{Q}} + \frac{6LN\sigma^2}{\sqrt{Q}T(\Gamma^*)^2} \tag{30}$$

$$+ \frac{2L^2NG}{Q\Gamma^*} + \frac{3L^2\delta^2N}{\Gamma^*}\cdot\frac{1}{Q}\sum_{q=1}^{T}\mathbb{E}|\theta_q|^2$$

$$= \frac{G_0}{\sqrt{Q}} + + \frac{V_0}{T\sqrt{Q}} + \frac{H_0}{Q} + \frac{I_0}{\Gamma^*}\cdot\frac{1}{Q}\sum_{q=1}^{Q}\mathbb{E}\|\theta_q\|^2, \tag{31}$$

where we introduce constants $G_0 = 4\mathbb{E}[F(\theta_0)]$, $V_0 = 6LN\sigma^2/(\Gamma^*)^2$, $H_0 = 2L^2NG/\Gamma^*$, and $I_0 = 3L^2\delta^2N$. This completes the proof of Theorem 1.

Finally, for non-IID data distribution, we plug the two upperbounds into Eq.(27) and re-arrange the terms. We follow a similar procedure and choose learning rate $\gamma \le 1/\sqrt{TQ}$ and $\gamma \le 1/(6LT)$. It is straightforward to show that for non-IID data distribution:

$$\frac{1}{Q}\sum_{q=1}^{Q}\mathbb{E}\|\nabla F(\theta_q)\|^2 \le \frac{G_1}{\sqrt{TQ}} + \frac{V_0}{\sqrt{Q}} + \frac{I_0}{\Gamma^*}\cdot\frac{1}{Q}\sum_{q=1}^{Q}\mathbb{E}\|\theta_q\|^2, \tag{32}$$

where $G_1 = 4\mathbb{E}[F(\theta_0)] + 6LK\sigma^2$ is a different constant. This completes the proof of Theorem 2.

## B  Experimental Details

### B.1  Experiment Setup

The code implementation is open sourced and can be found at

Github Link(Link anonymized, see supplementary materials for code and other tools).

In this experimental section we evaluate different pruning techniques from state-of-the-art designs and verify our proposed theory under unifying pruning framework using two datasets.

Unless stated otherwise, the accuracy reported is defined as

$$\frac{1}{n}\sum_i p_i \sum_j \text{Acc}(f_i(x_j^{(i)}, \theta_i \odot m_i), y_j^i))$$

averaged over three random seeds with same random initialized starting $\theta_0$. Some key hyperparameters includes total training rounds $Q = 100$, local training epochs $T = 5$, testing batch size $bs = 128$ and local batch size $bl = 10$. Momentum for SGD is set to 0.5. standard batch normalization is used.

We focus on three points in our experiments: (i) the general coverage of federated learning with heterogeneous models by pruning (ii) the impact of coverage index $\Gamma_{min}$ (iii) the impact of mask error $\delta$.

We examine the theoretical results on the following three commonly-used image classification datasets: MNIST with a shallow multilayer perception (MLP), CIFAR-10 with Wide ResNet28x2, and CIFAR100 with Wide ResNet28x8. The first setting where using MLP models is closer to the theoretical assumptions and settings, and the latter two settings are closer to the real-world application scenarios. We prepare $N = 100$ workers with IID and non-IID data with participation ratio $c = 0.1$ which will include 10 random active clients per communication round. For IID data, we follow the design of balanced MNIST by previous research, and similarly obtain balanced CIFAR10. For non-IID data, we obtained balanced partition with label distribution skewed, where the number of the samples on each device is up to at most two out of ten possible classifications.

### B.2  Pruning and submodel extraction Techniques

In the paper we select 4 pruning techniques as baselines and we elaborate the details of them. Let $P_m = \frac{\|m\|_0}{|\theta|}$ be the sparsity of mask $m$, e.g.,$P_m = 75\%$ for a model when 25 % of its weights are

pruned, and M is the number of the parameters in the model. Then a mask for weights pruning can be defined as:

$$m_i = \begin{cases} 1 & \text{, if } argsort(\theta[i]) < P_m * M \\ 0 & \text{, otherwise} \end{cases}, i \in M \tag{33}$$

where N is the total number of neurons in the network, and fixed subnetwork:

$$m_i = \begin{cases} 1 & \text{, if } i < P_m * M \\ 0 & \text{, otherwise} \end{cases}, i \in M \tag{34}$$

where M is the total number of parameters in the network.

Note in adaptive pruning such mask is subject to change after each round of global aggregation.

An illustration of those pruning techniques can be found in figure.

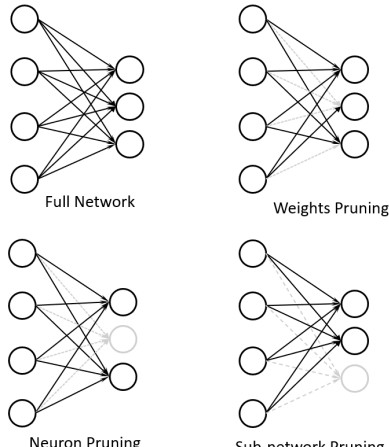

Figure 1: Illustration of pruning techniques used in this paper

## B.3 Evaluation Metrics

We use global model accuracy as our evaluation metrics. Specifically, global model accuracy is defined as the aggregated central server model accuracy on the test set. Local accuracy and other test and model details (e.g. FLOPs, model reduction ratio, etc.) can be found in the appendix. For all 3 datasets, we report the correct classification accuracy. Unless stated otherwise, the accuracy reported in this paper is defined as $\frac{1}{n} \sum_i p_i \sum_j \text{Acc}(f_i(x_j^{(i)}, \theta_i \odot m_i), y_j^i))$ averaged over three random seeds with the same random initialized starting $\theta_0$, conducted on 4 NVIDIA RTX2080 GPUs.

## C   More Results on MNIST dataset

In this section we present more supplementary experimental results on MNIST dataset as it's more close to our theoretical assumptions. Specifically, we present the training progress in respect of global loss and accuracy for selected pruning techniques.

### C.1   Change of Notations

In the main paper we use code name for simplicity of notation and better understanding. Here we present the results with their detailed settings.

For a full model without pruning it can be described as $\mathbb{P}_1(\theta) = \{S_1, S_2, S_3, S_4\}$, where

$$m_i = 1 \text{ if } \theta_i \in \{S_1 \cup S_2 \cup S_3 \cup S_4\} \text{ otherwise } m_i = 0$$

.

Similarly we have another 6 pruning polices as follows:

$$\mathbb{P}_2(\theta) = \{S_1, S_3, S_4\}$$
$$\mathbb{P}_3(\theta) = \{S_1, S_2, S_4\}$$
$$\mathbb{P}_4(\theta) = \{S_1, S_2, S_3\}$$

$$\mathbb{P}_5(\theta) = \{S_2, S_3\}$$
$$\mathbb{P}_6(\theta) = \{S_1, S_3\}$$
$$\mathbb{P}_7(\theta) = \{S_1, S_2\}$$

And we further denote a local client with its pruning policy, as an example, the case optimized medium model reduction uses 4 local clients with full models, 4 local clients with pruned models using pruning policy $\mathbb{P}_4$, 1 local client with pruned models using pruning policy $\mathbb{P}_2$ and 1 local client with pruned models using pruning policy $\mathbb{P}_3$, then we denote its code name as "1111234444" for simpler notation. Note that we continue to use code name "FedAvg" as a baseline rather than "1111111111". For the rest of the appendix we continue using such notations for denoting its model reduction policy settings.

| codename | 1 | 0.75 | 0.5 | PARAs | FLOPs | $\Gamma_{min}$ | %PARA | %FLOPS | IID | Non-IID | |
| --- | --- | --- | --- | --- | --- | --- | --- | --- | --- | --- | --- |
| | | | | | | | | | Accuracy | Global | Local |
| 1111111111 | 10 | | | 159010 | 158800 | 10 | 1.00 | 1.00 | 98.045 | 93.59 | 93.82 |
| 1111114444 | 6 | 4 | | 143330 | 143120 | 6 | 0.90 | 0.90 | 98.18 | 95.15 | 95.49 |
| 1111144447 | 5 | 4 | 1 | 135490 | 135280 | 5 | 0.85 | 0.85 | 97.51 | 89.13 | 89.29 |
| 1111223344 | 4 | 6 | | 135490 | 135280 | 8 | 0.85 | 0.85 | 98.32 | 95.48 | 95.82 |
| 1111234444 | 4 | 6 | | 135490 | 135280 | 6 | 0.85 | 0.85 | 98.39 | 95.45 | 95.96 |
| 1111113477 | 6 | 2 | 2 | 135490 | 135280 | 7 | 0.85 | 0.85 | 96.72 | 91.27 | 91.57 |
| 1111234567 | 4 | 3 | 3 | 123730 | 123520 | 7 | 0.77 | 0.77 | 96.73 | 88.99 | 88.90 |
| 1111444444 | 4 | 6 | | 135490 | 135280 | 4 | 0.85 | 0.85 | 97.85 | 89.13 | 89.29 |
| 1111444477 | 4 | 4 | 2 | 127650 | 127440 | 4 | 0.80 | 0.80 | 96.9 | 93.02 | 93.12 |
| 1111556677 | 4 | | 6 | 111970 | 111760 | 6 | 0.70 | 0.70 | 95.5 | 80.07 | 79.34 |
| 1114556677 | 3 | 1 | 6 | 108050 | 107840 | 5 | 0.67 | 0.67 | 95.80 | 79.30 | 79.75 |
| 1234556677 | 1 | 3 | 6 | 100210 | 100000 | 5 | 0.63 | 0.62 | 95.31 | 81.66 | 81.64 |
| 1455666777 | 1 | 1 | 8 | 92370 | 92160 | 3 | 0.58 | 0.58 | 94.79 | 79.15 | 79.08 |
| 2233445677 | 0 | 6 | 4 | 104130 | 103920 | 5 | 0.65 | 0.65 | 95.95 | 81.27 | 81.17 |
| 1444777777 | 1 | 3 | 6 | 92370 | 92160 | 6 | 0.65 | 0.65 | 95.10 | 72.19 | 71.64 |

Table 2: Results For Weights Pruning on MNIST

| codename | 100% | 75% | 50% | PARAs | FLOPs | $\Gamma_{min}$ | %PARA | %FLOPS | IID | Non-IID | |
| --- | --- | --- | --- | --- | --- | --- | --- | --- | --- | --- | --- |
| | | | | | | | | | Accuracy | Global | Local |
| 1111111111 | 10 | | | 159010 | 158800 | 10 | 1.00 | 1.00 | 97.67 | 94.12 | 94.45 |
| 1111114444 | 6 | 4 | | 143110 | 142920 | 6 | 0.9 | 0.90 | 97.76 | 92.33 | 92.55 |
| 1111144447 | 5 | 4 | 1 | 135160 | 134980 | 6 | 0.85 | 0.85 | 97.34 | 93.79 | 93.92 |
| 1111444444 | 4 | 6 | | 135160 | 134980 | 4 | 0.85 | 0.85 | 97.62 | 92.05 | 92.33 |
| 1111444477 | 4 | 4 | 2 | 127210 | 127040 | 4 | 0.80 | 0.80 | 97.32 | 92.67 | 92.95 |
| 1111444777 | 4 | 3 | 3 | 123235 | 123070 | 4 | 0.77 | 0.77 | 97.35 | 91.34 | 91.73 |
| 1111777777 | 4 | | 6 | 111310 | 111160 | 4 | 0.70 | 0.70 | 97.18 | 93.6 | 93.48 |
| 1114777777 | 3 | 1 | 6 | 107335 | 107190 | 3 | 0.67 | 0.67 | 97.12 | 93.7 | 93.57 |
| 1444777777 | 1 | 3 | 6 | 99385 | 99250 | 1 | 0.62 | 0.62 | 97.01 | 90.74 | 90.57 |
| 1477777777 | 1 | 1 | 8 | 91435 | 91310 | 1 | 0.57 | 0.57 | 96.88 | 90.73 | 90.67 |

Table 3: Results For Fixed Sub-network on MNIST

## C.2 More Results

### C.2.1 Case for IID data

We present the full results of training for IID case in Fig 2 - 3

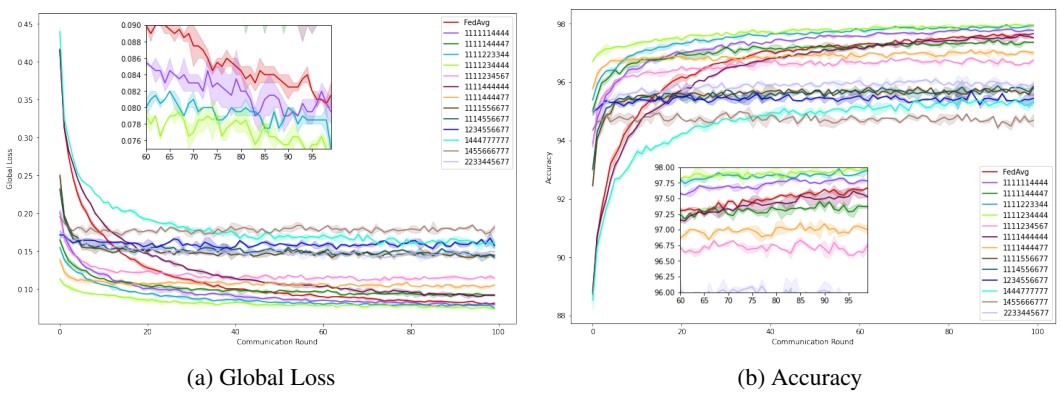

(a) Global Loss

(b) Accuracy

Figure 2: Results on Weights Pruning on MNIST IID

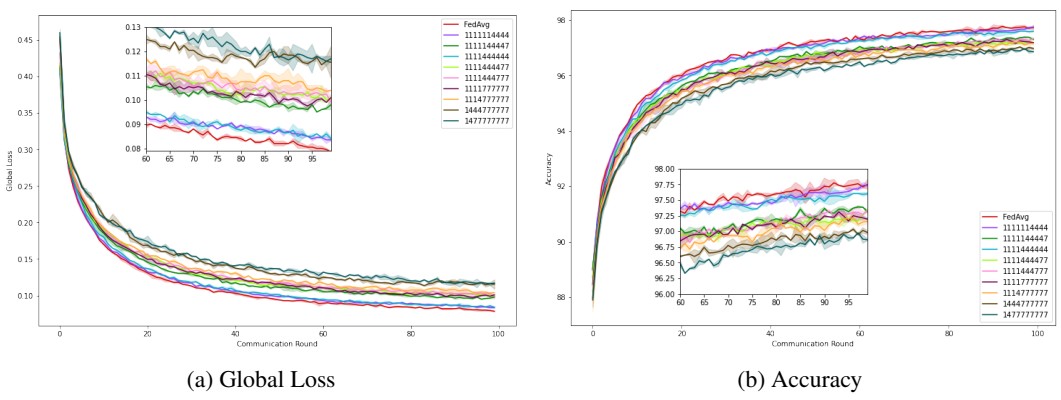

(a) Global Loss

(b) Accuracy

Figure 3: Results on Fixed Sub-network on MNIST IID

### C.2.2 Case for non-IID data

We present the full results of training for non-IID case in Fig 4 - 5

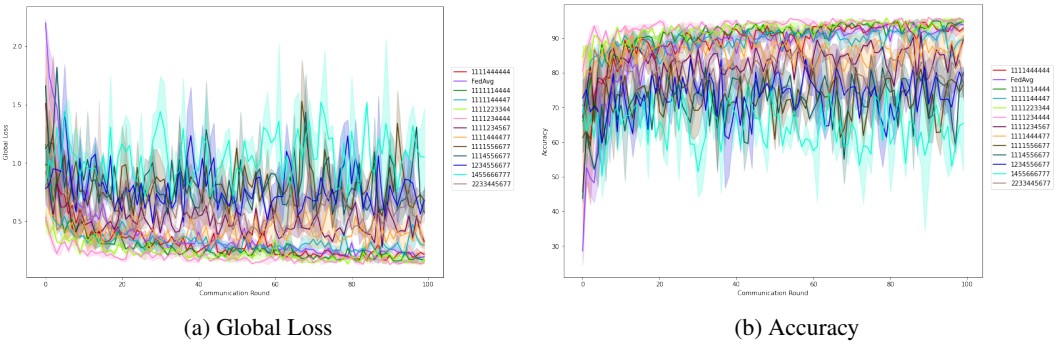

(a) Global Loss

(b) Accuracy

Figure 4: Results on Weights Pruning on MNIST non-IID

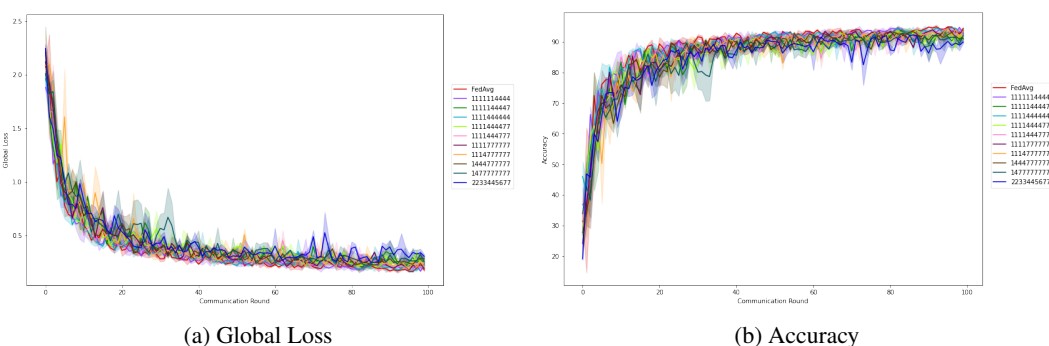

(a) Global Loss    (b) Accuracy

Figure 5: Results on Fixed Sub-network on MNIST non-IID