# OpenReview forum: "Every Parameter Matters: Ensuring the Convergence of Federated Learning with Dynamic Heterogeneous Models Reduction"
_NeurIPS.cc/2023/Conference — NeurIPS 2023 poster_

### Official Review · Reviewer_dR1n · 2023-07-03

**Soundness:** 4 excellent
**Presentation:** 4 excellent
**Contribution:** 4 excellent
**Rating:** 7
**Confidence:** 4

**Summary:**

This work establishes a general convergence analysis for a heterogeneous federated learning (FL) algorithm that trains a shared global model using a sequence of time-varying and client-dependent local models. In particular, this work establishes sufficient conditions for the convergence of such a heterogeneous FL algorithm to the neighborhood of a stationary point of the standard FL. Based on the theoretical results, the authors propose practical suggestions for designing heterogeneous FL algorithms and conduct thorough experiments to support their claims.

**Strengths:**

Overall, this paper is well-written. It is the first to provide a general convergence result to the neighborhood of a stationary point for the standard federated learning (FL) framework, specifically for a heterogeneous FL algorithm that trains a shared global model through a sequence of time-varying and client-dependent local models. This result provides a convergence guarantee for several previously proposed heterogeneous FL algorithms. The optimality gap in the result depends on two important parameters: the minimum coverage index and the model reduction noise. These parameters offer valuable insights for designing practical heterogeneous FL algorithms. The experimental results align well with the theoretical findings presented in the paper.

**Weaknesses:**

The last term on the right-hand side of the inequality in Theorem 1 (and similarly for Theorem 2) corresponds to the average of the norms of the global parameters encountered during the optimization process. This term is not necessarily small in a straightforward manner.

**Questions:**

Can the authors provide additional discussions regarding the last term in Theorem 1 (and Theorem 2)? Specifically, it would be helpful to explore whether this term is tight or under what circumstances it can be considered small.

---

> ### Author Rebuttal · Authors · 2023-08-08
>
> We thank the reviewer for their constructive and insightful comments and suggestions. Following are our responses to your questions and concerns.
>
> > Q1. The last term on the right-hand side of the inequality in Theorem 1 (and similarly for Theorem 2) corresponds to the average of the norms of the global parameters encountered during the optimization process. This term is not necessarily small in a straightforward manner.
>
> * It is true that it might be hard to directly quantify the term  $E\Vert \theta_q\Vert^2$, since it depends on the exact FL problem setup. We note that this term $E\Vert \theta_q\Vert^2$ is bounded and should be examined together with its coefficients in the convergence bound, i.e., $\frac{\delta^2}{\Gamma^*} E\Vert \theta_q\Vert^2$. In particular, considering the coefficient $\delta^2$ together with $E\Vert \theta_q\Vert^2$, we have $E\Vert \delta \theta_q\Vert^2$, which relates to the pruning noise introduced in Assumption 2 and is equivalent to the bounded model reduction noise, i.e., $\|\theta_{q}-\theta_{q} \odot m_{q,n}\|^{2} \leq \delta^{2}\|\theta_{q}\|^{2}$. We also note that it is common to have a bounded model norms term in heterogeneous federated learning such as [1][2]; or a similar term of $|w_1-w_*|^2$ in works e.g. [3]. Our analysis shows that to reduce this term $\frac{\delta^2}{\Gamma^*} E\Vert \theta_q\Vert^2$, we can either increase $\Gamma_{min}$ (by designing pruning strategies) or decrease mask noise $\delta$ (by adjusting pruning level). We will add these discussions to the revised paper.
>
> >Q2. Can the authors provide additional discussions regarding the last term in Theorem 1 (and Theorem 2)? Specifically, it would be helpful to explore whether this term is tight or under what circumstances it can be considered small.
>
> * We will add more discussion on this term in the revision. Although it is hard to directly tell how large this $E\Vert \theta_q\Vert^2$ is, our evaluation results demonstrate how this term affects learning performance. For instance, in the main results (Table 1 in main paper) we can compare “Pruning-Greedy” and “Pruning-Optimised”: the former has a larger $E\Vert \theta_q\Vert^2$ and the latter has a larger $\Gamma_{min}$, and as a result the latter one comes with a lower loss and converges with a faster convergency rate. Similar results can be found in other cases in the main results (e.g. by comparing  “Pruning-Medium (Greedy)” and “Pruning-Medium-Optimised” in Fig2 (a,b)) and additional results in the appendix. We will add these explanations and more insights in the revised paper.
>
> [1] Jiang, Zhida, et al. "Fedmp: Federated learning through adaptive model pruning in heterogeneous edge computing." 2022 IEEE 38th International Conference on Data Engineering (ICDE). IEEE, 2022.
>
> [2] Xiaolong Ma, et al. "Effective Model Sparsification by Scheduled Grow-and-Prune Methods." International Conference on Learning Representations. 2022.
>
> [3]Xiang Li, et al. "On the Convergence of FedAvg on Non-IID Data." International Conference on Learning Representations. 2020.

---

> > ### Comment · Reviewer_dR1n · 2023-08-14
> >
> > Thanks for the rebuttal and the discussion the authors try to add in the future. I will take it into consideration during the discussion phase with AC.

---

### Official Review · Reviewer_mRuV · 2023-07-05

**Soundness:** 3 good
**Presentation:** 3 good
**Contribution:** 2 fair
**Rating:** 5
**Confidence:** 5

**Summary:**

This paper studies federated learning with heterogeneous client models and non-iid client data. By assuming that the client models are pruned versions of a common global model and using the notion of minimum covering index, the paper provides the convergence of FedAvg under client model pruning. Theoretical results show that pruning techniques that more evenly update the parameters and result in smaller model distortion are preferable to aggressive pruning. Such result is also verified through the numerical results on MNIST and Cifar-10, and Cifar-100 datasets.


**Strengths:**

Novelty: this paper provides a first convergence analysis to FedAvg with model pruning. The theoretical proof is sound and clear.

Clarity: the paper thoroughly discusses the convergence result in theorems 1 and 2 and verifies the results through numerical experiments.

**Weaknesses:**

1. Term $E\Vert \theta_q\Vert^2$ in the bound. This term appears in all theorems and lemmas in the main paper. However, it is unclear how large this term can be. Neither theoretical discussion nor numerical justification of this term is provided. This undermines the strength of the theoretical results.

2. The connection between the theorem to the general convergence result of FedAvg under the same assumption. When $\delta = 0, \Gamma_{min} = T$, the convergence should recover the rate of standard FedAvg. However, such a discussion is missing, weakening the paper's clarity.

3. The connection between the proof of pruning and lossy compression. Assumption 2 takes the standard assumption of model compression in compressed FL (e.g., [19, 21, 22]). Yet there is a critical difference in the minimum covering index. The connection and difference between the proof technique should be discussed.

**Questions:**

Please address the above weaknesses.

**Limitations:**

The authors have discussed the limitation of the paper, that partial client participation is not considered, and how the theoretical result guides the design of an optimal model pruning strategy is also limited.

---

> ### Author Rebuttal · Authors · 2023-08-08
>
> Thank you for the constructive comments. We will follow these helpful comments in our revised version. Following are our responses to your questions and concerns.
>
> > Q1. Term $E\Vert \theta_q\Vert^2$  in the bound. This term appears in all theorems and lemmas in the main paper. However, it is unclear how large this term can be. Neither theoretical discussion nor numerical justification of this term is provided. This undermines the strength of the theoretical results.
>
> * Indeed it is hard to directly quantify the term  $E\Vert \theta_q\Vert^2$, since it depends on the exact FL problem setup. We note that this term $E\Vert \theta_q\Vert^2$ should be examined together with its coefficients in the convergence bound, i.e., $\frac{\delta^2}{\Gamma_{min}} E\Vert \theta_q\Vert^2$. In particular, considering the coefficient $\delta^2$ together with $E\Vert \theta_q\Vert^2$, we have $E\Vert \delta \theta_q\Vert^2$, which relates to the pruning noise introduced in Assumption 2 and is equivalent to the bounded model reduction noise, i.e., $\|\theta_{q}-\theta_{q} \odot m_{q,n}\|^{2} \leq \delta^{2}\|\theta_{q}\|^{2}$. We also note that it is common to have such a bounded model norms term in heterogeneous federated learning, such as [1][2]; or a similar term of $|w_1-w_*|^2 $  in works e.g. [3]. Our analysis shows that to reduce this term $\frac{\delta^2}{\Gamma_{min}} E\Vert \theta_q\Vert^2$, we can either increase $\Gamma_{min}$ (by designing pruning strategies) or decrease mask noise $\delta$ (by adjusting pruning level). We will add these discussions to the revised paper.
>
>
> * Although it is hard to directly tell how large this $E\Vert \theta_q\Vert^2$ is, our evaluation results demonstrate how this term affects learning performance. For instance, in the main results (Table 1 in main paper) we can compare “Pruning-Greedy” and “Pruning-Optimised”: the former has a larger $E\Vert \theta_q\Vert^2$ and the latter has a larger $\Gamma_{min}$, and as a result the latter one comes with a lower loss and converges at a faster convergence rate. Similar results can be found in other cases in the main results (e.g. by comparing  “Pruning-Medium (Greedy)” and “Pruning-Medium-Optimised” in Fig2 (a,b)) and additional results in the appendix.
>
>
>
>
> > Q2. The connection between the theorem to the general convergence result of FedAvg under the same assumption. When $\delta = 0, \Gamma_{min} = T$, the convergence should recover the rate of standard FedAvg. However, such a discussion is missing, weakening the paper's clarity.
>
> * This is a good point.  While $\delta = 0, \Gamma_{min} = T$ reduces the system model to standard FedAvg, our proof approach here is quite different due to additional steps (i.e., Lemma 1,2,3 in appendix) that are required to bound the training of heterogeneous local models, their differences at the end of each round, and the impact on global aggregation. Thus, we may not exactly recover the same convergence result as FedAvg. Nevertheless, as shown in Theorems 1 and 2: “We prove that heterogeneous FL algorithms satisfying certain sufficient conditions can indeed converge to a neighborhood of a stationary point of standard FL (with a small optimality gap that is characterized in our analysis)”.  When $\delta = 0, \Gamma_{min} = T$, the radius of the error ball (i.e., the last term in the convergence bounds) indeed becomes zero, implying convergence to standard FedAvg. This discussion will be added to in the revised paper.
>
>
> > Q3. The connection between the proof of pruning and lossy compression. Assumption 2 takes the standard assumption of model compression in compressed FL (e.g., [19, 21, 22]). Yet there is a critical difference in the minimum covering index. The connection and difference between the proof technique should be discussed.
>
> * There are three key differences: (1) In this paper, we introduced the concept of minimum coverage index for the first time, where we show that only model compression alone is not enough to allow a unified convergence analysis/framework for heterogeneous federated learning. Minimum coverage index, together with pruning/compression noises, determines convergence in heterogeneous FL.  (2) Our results show that heterogeneous FL algorithms satisfying certain sufficient conditions can indeed converge to a neighborhood of a stationary point of standard FL. This is a stronger result as it shows convergence to standard FL, rather than simply conversing to somewhere. A minimum coverage index of $\Gamma_{min}=0$ means that the model would never be updated, which is meaningless even if it still converges. (3) In terms of the proof techniques, additional steps (i.e., Lemma 1,2,3 in appendix) are required to bound the training of heterogeneous local models, quantify their differences at the end of each round, and analyze the impact on global aggregation. These make the proof much more complicated than convergence analysis of standard FL.  We will add more discussions on these in the revised paper.
>
>
> [1] Jiang, Zhida, et al. "Fedmp: Federated learning through adaptive model pruning in heterogeneous edge computing." 2022 IEEE 38th International Conference on Data Engineering (ICDE). IEEE, 2022.
>
> [2] Xiaolong Ma, et al. "Effective Model Sparsification by Scheduled Grow-and-Prune Methods." International Conference on Learning Representations. 2022.
>
> [3]Xiang Li, et al. "On the Convergence of FedAvg on Non-IID Data." International Conference on Learning Representations. 2020.

---

> ### Author Response · Authors · 2023-08-18
>
> Dear reviewer mRuV,
>
> We would like to thank you again for the time you dedicated to reviewing our paper and your valuable comments. We believe that we have addressed your concerns. Since the end of the discussion period is approaching and we have not heard back from you yet, we would appreciate it if you kindly let us know of any other concerns you may have, and if we can be of any further assistance in clarifying any other issues.
>
> Thanks and sincerely,
>
> Authors

---

> > ### Comment · Reviewer_mRuV · 2023-08-21
> >
> > I thank the author for their response, and those clarifications should be added in the revised manuscript. Other than that ,I think this is a paper above the accept line. I will take the response into consideration during the discussion phase with AC.

---

### Official Review · Reviewer_oYBg · 2023-07-06

**Soundness:** 3 good
**Presentation:** 3 good
**Contribution:** 3 good
**Rating:** 5
**Confidence:** 2

**Summary:**

This paper focuses on the cross-device federated learning setting. The authors introduce a general theoretical framework for analyzing FedAvg with masks on local pruned models. This analysis is particularly valuable in establishing the convergence of federated schemes when the global model is distributed across multiple edge clients. Furthermore, the paper includes numerical illustrations of their algorithm, providing practical insights on its performance.

**Strengths:**

* The paper is well-written and easy to understand, but with few points that might be improved (see "Questions" part).

* The paper conducts a thorough theoretical investigation on federated learning with reduced-size models. Specifically, the authors provide novel tools to study the convergence of FL pruned models.

**Weaknesses:**

* Although interesting, some parts may seem to have been rushed. For example, assumptions and lemmas should be referred with ref{}. Further, I also spotted a few errors on the supplement, even though I haven't read the supplementary in detail. It's important to review and correct these typos.


**Questions:**

The article is interesting, could you address the following points so that I can raise my score?

* It is unclear whether $\Gamma_q^{(i)}$ is defined or if it refers to $N_{q}^{(i)}$ mentioned on line 177. Similarly, does $\Gamma^*$ denote the same quantity as $\Gamma_{min}$?

* The proposed update in the equation on line 134 is not exactly the same implemented in the "Update.py" script. I think the mask is not applied to the gradient.

* I haven't looked at the supplementary paper, but it seems to me that an expectation is missing on Eq.(20).

* Is it necessary to use $\delta<1$ in assumption 2? Since I don't see $1/(1-\delta)$ in your bounds, I wonder if this constraint is useful.

**Requested Changes:**

* Line 59: "will it converge" inside Figure 1 legend.

* Line 75: client is denoted by $i$, then latter by $n$ at the end of page 2.

* Line 95: "models are to share".

* Lines 108: "works like Hermes can".

* Definition of $I_0$ in Theorem 1 and Theorem 2 is different, there is maybe an additional $\delta^2$.

* Some punctuation is missing, e.g., Eq. (9).

* Conclusion seems little heavy.

**Limitations:**

The authors did not address the potential negative societal implications of their research, but this does not seem critical for this particular theoretical and numerical study.

---

> ### Author Rebuttal · Authors · 2023-08-08
>
> Thank you for your comments and for confirming the contributions of our paper. We provide clarification to your questions and concerns as below.
>
> > Q1. It is unclear whether $ \Gamma_q^{(i)}$  is defined or if it refers to $N_{q}^{(i)}$  mentioned on line 177. Similarly, does $\Gamma^*$  denote the same quantity as $\Gamma_{min}$?
>
> * Yes, $ \Gamma_q^{(i)}$ is defined according to Eq(8) on line 177. Essentially $\Gamma^*$ and $\Gamma_{min}$ are the same thing. $\Gamma_{min}$  is introduced as a main concept of the paper, and  $\Gamma^*$ is used during the derivation for simplicity. We will clarify this in the revision.
>
> > Q2. The proposed update in the equation on line 134 is not exactly the same implemented in the "Update.py" script. I think the mask is not applied to the gradient.
>
> * Thanks for checking our code implementation! In this proof-of-concept experiment, we did not apply masks to the gradients, as this would cause an error if we directly modify the gradients since PyTorch autograd framework is responsible for handling gradients. Instead, for each batch of training, we directly apply the mask to the weights and biases before calculating the loss and after the back-propagation (e.g. the use of “get_sub_paras” helper function at Line 22 in main_fed_20N_AVGALL.py). This is equivalent to applying the mask to the gradients and has been used as a workaround in previous work for more easily obtaining proof-of-concept implementations.
>
> > Q3. I haven't looked at the supplementary paper, but it seems to me that an expectation is missing on Eq.(20).
>
> * Thanks for examining the appendix section and pointing this out, indeed there should be an expectation sign in Eq.(20). We have carefully proofread the supplementary document again and fixed the typos.
>
> > Q4. Is it necessary to use $\delta<1$  in assumption 2? Since I don't see  $1/(1-\delta)$ in your bounds, I wonder if this constraint is useful.
>
> * It is necessary and useful to have this condition. In fact, $\delta<1$ holds by definition and directly follows from Assumption 2. Since the local model is extracted from the global model, it has to be smaller than the global model (as some parameters are pruned). According to Eq.(11) to quantify the difference/noise resulting from applying a mask, $\delta$ must be smaller than 1 by definition. That’s why we listed it in the assumption. We will add a footnote to clarify this in the revised paper.
>
> > Definition of $I_0$  in Theorem 1 and Theorem 2 is different, there is maybe an additional $\delta^2$
>
> * Thanks for pointing this typo out. The definition of $I_0$ in Theorem 2 should be exactly the same as that in Theorem 1 (thus the additional $\delta^2$ is absorbed into $I_0$ in Theorem 2 as well). This is a type introduced when we were trying to simplify the auxiliary variables in both theorems. Full equations and their derivations are presented in the appendix.
>
>
> > Requested Changes:
> Some punctuation is missing, e.g., Eq. (9).
> Line 59: "will it converge" inside Figure 1 legend.
> Line 75: client is denoted by $i$, then latter by $n$  at the end of page 2.
> Line 95: "models are to share".
> Lines 108: "works like Hermes can".
> Conclusion seems little heavy.
> * Thanks for pointing out the typos, grammar, format issues, and writing suggestions, they will be addressed in the revision.

---

> ### Author Response · Authors · 2023-08-18
>
> Dear reviewer oYBg,
>
> We would like to thank you again for the time you dedicated to reviewing our paper and your valuable comments. We believe that we have addressed your concerns. Since the end of the discussion period is approaching and we have not heard back from you yet, we would appreciate it if you kindly let us know of any other concerns you may have, and if we can be of any further assistance in clarifying any other issues.
>
> Thanks and sincerely,
>
> Authors

---

### Official Review · Reviewer_9LDk · 2023-07-07

**Soundness:** 3 good
**Presentation:** 3 good
**Contribution:** 3 good
**Rating:** 6
**Confidence:** 4

**Summary:**

In this work, the authors provide a general theoretical framework to analyze the convergence of Federated training schemes conducted over local models of heterogeneous network structures. Such structures can usually be obtained through different model reduction methods, such as model pruning/sparsification or model extraction. The proposed framework introduces the minimum covering index concept to conduct the analysis, representing the number of local models concurrently updating the same set of parameter indices. The paper is well written, but a couple of clarifications are necessary.


**Strengths:**

Very well-written paper with clear objectives and contributions.

Generalized framework to encapsulate different model reduction algorithms.

The introduction of the minimum coverage index concept and its interplay with model reduction noise can lead to very promising and interesting insights.

**Weaknesses:**

Further elaboration is needed on terms and concepts used during the theoretical analysis.

Empirical evaluation needs improvement.

**Questions:**

From my understanding, your framework seems to consider reduced model structures when these structures are obtained when the global model is reduced (pruned). Will your analysis hold even in the case of local model reduction, i.e., client-side reduction, not server-side? Will the model reduction noise (assumption 2) and bounded gradient (assumption 3) still hold? Your analysis shows that local model training is performed over the entire network. How would that affect your analysis if you were to enforce training over the parameters of the reduced model (enforce masks during training)?

The proof outline does not clarify how the mask is constructed. Is it always static, or can it be dynamically changing? In PruneFL, masks can periodically increase to accommodate model regrowth. Similarly, FedDST[1] performs local model regrowth with a fixed pruning degree during model communication. Analogously, FedSparsify[2] performs progressive model reduction (progressive sparsification), where the global or local model is progressively pruned. Can your framework capture such dynamic mask construction? Please expand your related work and discuss whether your framework can also accommodate such dynamic pruning approaches.

The model settings through which the model reduction level is obtained in Table 1 are not clear. Can you please elaborate on what are the methods you used to perform the greedy pruning, pruning-optimized, static subnet subtraction, and homogeneous methods? Did you use structured or unstructured pruning? Random weight magnitude or based on magnitude? Can you report the noise reduction (percentage-wise) in the models learned at every model reduction level? Such an analysis could lead to key insights with respect to the index coverage vs. noise reduction tradeoff.

Can your analysis be extended to activations and/or neuron pruning as part of your future work? Also, have you considered extending your framework for asynchronous federated settings?

[1] Federated Dynamic Sparse Training: Computing Less, Communicating Less, Yet Learning Better Sameer Bibikar, Haris Vikalo, Zhangyang Wang, Xiaohan Chen. https://ojs.aaai.org/index.php/AAAI/article/view/20555/20314

[2] Federated Progressive Sparsification (Purge-Merge-Tune)+. Dimitris Stripelis, Umang Gupta, Greg Ver Steeg, Jose Luis Ambite. https://openreview.net/pdf?id=GLQqPTRrQMx

**Limitations:**

No.

---

> ### Author Rebuttal · Authors · 2023-08-08
>
> Thanks for your review!
>
> > Q1. From my understanding, your framework seems to consider reduced model structures when these structures are obtained when the global model is reduced (pruned). Will your analysis hold even in the case of local model reduction ... How would that affect your analysis if you were to enforce training over the parameters of the reduced model?
>
> We would like to clarify that our framework indeed considers local model training over the reduced networks (i.e., the masks are always enforced during the training of local models). It is shown in Eq.(2) at line 131 and the equation at line 134 how the local models are obtained by pruning and then trained as reduced networks. A pseudocode example is provided in Algorithm 1 (page 1) in the appendix, showing the training of local models with fixed masks and their aggregation to the global model. If we understand the comment correctly, this is the client-side reduction and reduced local model training, as the reviewer pointed out. This problem formulation makes convergence difficult to analyze, which is the main contribution of this paper. Our analysis shows that under Assumptions 2  and 3 (both of which relate to reduced local models), convergence (to standard FL) will stand for as long as certain conditions are met, that is, every parameter is included and updated at least once throughout the training, which is the key idea of this paper: every parameter matters. Further, our experiment section results are generated in a way that local model training is done on reduced-size models.
>
> > Q2. The proof outline does not clarify how the mask is constructed. Is it always static, or can it be dynamically changing? .... Can your framework capture such dynamic mask construction? Please expand your related work and discuss whether your framework can also accommodate such dynamic pruning approaches.
>
> Our paper establishes convergence conditions in the general form, which apply to both static and dynamically changing masks. With dynamically changing masks, we denote the reduced models/networks as  $\theta_{q,n,t}$, which means that the model structure (with its corresponding mask) can change between each round of communications $q$ and during each local training epoch $t$, and can be different from other local clients. We show that as long as the dynamic heterogenous FL framework can be framed as the setting above, our convergence analysis in this paper applies. Thus, our results establish the general convergence condition covering cases of static model pruning, dynamic model reduction, and grow-and-prune type of training, e.g. [1-4]. However, indeed the current description of mask generation can be improved by expanding to more related works for better understanding. We will make such changes in the revision as the reviewer suggested.
>
> > Q3. The model settings through which the model reduction level is obtained in Table 1 are not clear. Can you please elaborate on what are the methods you used to perform the greedy pruning, pruning-optimized, static subnet subtraction, and homogeneous methods?  .....  Such an analysis could lead to key insights with respect to the index coverage vs. noise reduction tradeoff.
>
> Due to page limitations, we summarized the key findings in the main paper and presented detailed settings of each model reduction method in the appendix. Specifically, we listed how local models are generated (how we define greedy pruning, pruning-optimized, static subnet subtraction, and how masks are generated, etc) and their respective coverage index and each of their reduction levels in percentage-wise (which leads to the initial model reduction noise).
>
> For instance, we explained that "Pruning-Optimised" works at a low model reduction level and is made up with 10 local models consisting of 4 full-size models, 6 reduced-size models at 75% size of the global model that covers 3 different regions (with each setting applied to 2 models) as  {$S_{1},S_{3},S_{4}$},  {$S_{1},S_{2},S_{4}$},  {$S_{1},S_{2},S_{3}$}, where $S_1$ represents the region consists of the top 25% largest-magnitude parameters and $S_4$ the smallest, and   {$S_{1},S_{2},S_{3},S_{4} $} as the full model. We considered both structured and unstructured pruning including unstructured weights pruning, structured neuron pruning, and leading subnet extraction. We will add a brief summary to further clarify this in the revised paper.
>
> > Q4. Can your analysis be extended to activations and/or neuron pruning as part of your future work? Have you considered extending your framework for asynchronous federated settings?
>
> The current setting of model reduction considers both structured and unstructured model reduction, which includes neuron pruning. In fact, in the experiment section and appendix, we have results for neuron pruning. In Table 1 of the main paper, we used “pruning” to demonstrate unstructured pruning and “static subnet subtraction” to demonstrate structured neuron pruning (continuous neuron pruning), and due to page limits, we have in the appendix results for (unstructured) neuron pruning in table 2 and table 3 and their illustrations in Figure 1.
>
> We agree extending the framework to asynchronous federated settings would be an interesting direction for future work. We will include some discussions in the revised paper.
>
> [1] Bibikar, Sameer, et al. "Federated dynamic sparse training: Computing less, communicating less, yet learning better." Proceedings of the AAAI Conference on Artificial Intelligence. Vol. 36. No. 6. 2022.
>
> [2]  Stripelis, Dimitris, et al. "Federated progressive sparsification (purge, merge, tune)+." arXiv preprint arXiv:2204.12430 (2022).
>
> [3] Tao Lin, et al. "Dynamic Model Pruning with Feedback." International Conference on Learning Representations. 2020.
>
> [4] Alam, Samiul, et al. "Fedrolex: Model-heterogeneous federated learning with rolling sub-model extraction." Advances in Neural Information Processing Systems 35 (2022): 29677-29690.

---

> > ### Comment · Reviewer_9LDk · 2023-08-16
> >
> > Thank you for clarifying that the masks are enforced during local model training. It was not evident in the original text; you might need to state this explicitly. I also appreciate your effort in explaining the different types of model reduction you considered in your evaluation and how your analysis encapsulates various pruning methods. Overall, the authors have addressed all of my concerns. My score remains the same.

---

> > > ### Author Response · Authors · 2023-08-18
> > >
> > > Dear reviewer 9LDk,
> > >
> > > We would like to thank you again for the time you dedicated to reviewing our paper and your valuable comments that will further improve the clarity of the paper.
> > >
> > > We are happy to see that our response has addressed all of your concerns.
> > >
> > > Thanks a lot again and with sincerest best wishes,
> > >
> > > Authors

---

### Author Rebuttal · Authors · 2023-08-10

Dear reviewers, thank you all again for your valuable time and positive suggestions that will definitely make our paper stronger. We have provided each reviewer with our responses in a Q&A format. We are happy to answer further questions if any.

---

### Decision · Program_Chairs · 2023-09-21

**Decision:**

Accept (poster)

**Comment:**

The paper Introduces a general theoretical framework for analyzing the convergence of Federated training schemes over local models of heterogeneous network structures. The paper will be interesting for the community since it presents a more generalized federated pruning framework. The paper is technically solid and offers convergence guarantees for various heterogeneous federated learning algorithms, while numerical illustrations provide practical insights on performance. Experimental results align well with theoretical findings.

That said, the reviewers point out a number of things that need to be clarified in the final version of the paper. Please take into considerations the comments made by reviewers that will improve the quality of the paper.